# $f$-Divergence Self-Play for Tabular Anomaly Detection via Large Language Models

**Hoang Tran Vuong** [1]  **Linh Ngo Van** [1]  **Dang Nguyen** [2]  **Thin Nguyen** [2]  **Phuoc Nguyen** [2]  **Mehrtash Harandi** [3]  **Trung Le** [3]

## Abstract

Anomaly detection in tabular data poses significant challenges due to heterogeneous feature types—mixing numerical, categorical, and textual attributes, which complicate learning meaningful representations of normality. Recent work has applied large language models (LLMs) to this problem by serializing table rows as text sequences, yet these approaches rely on one-shot supervised fine-tuning that offers limited signal to tighten the model's description of normality. We propose DiSPaT, a self-play fine-tuning framework that strengthens the model's understanding of normal data. Building on the theoretical foundation of $f$-divergence minimization, we derive a tight approximation connecting our training objective to reducing the distributional gap between real normal data and model-generated samples. DiSPaT operates through an alternating optimization: at each iteration, the current policy generates synthetic samples that serve as pseudo-anomalies, while a critic discriminator learns to distinguish these from real normal samples; this signal drives policy updates that progressively align the model distribution with the true normal-data distribution. Extensive experiments on diverse benchmarks demonstrate that DiSPaT consistently outperforms prior LLM-based methods, deep learning approaches, and classical unsupervised detectors for tabular anomaly detection.

## 1. Introduction

Anomaly detection (AD) aims to identify rare or abnormal instances that deviate from an underlying notion of "normality" (Hawkins, 1980; Chandola et al., 2009). This problem is central in high-stakes applications where anomalies may indicate fraudulent transactions, cybersecurity intrusions, safety issues, clinical outliers, or abnormal events in surveillance video (Ngai et al., 2011; Buczak & Guven, 2015; Vu et al., 2019; Fernando et al., 2021; Landauer et al., 2023). In practice, data in these domains are often tabular and heterogeneous: each record may mix numerical measurements, categorical attributes, and increasingly, free-form text with missing values and non-trivial cross-field dependencies. These characteristics make unsupervised detection difficult, as standard tabular AD pipelines typically assume fully structured features and either handle text only crudely or rely on heavy feature engineering that can remove useful semantics and miss complex interactions. Large language models (LLMs) have recently become an attractive foundation for mixed-type tabular AD because they can operate on a serialized representation of each row and naturally exploit the semantics of categorical values and raw text (Dinh et al., 2022; Hegselmann et al., 2023; Fang et al., 2024). Viewed as a serialized sequence, each row can be processed by an LLM in a unified way, allowing the model to exploit the semantics of categorical values and raw text with minimal preprocessing. However, applying LLMs to unsupervised tabular AD is non-trivial. Concretely, the serialization should respect basic tabular invariances (e.g., column order), numerical fields require representations that are robust to tokenizer-induced distortions, and likelihood-based scores should be calibrated to avoid systematic bias from variable-length text.

A foundational early work that first demonstrated the practicality of applying LLMs to tabular anomaly detection is AnoLLM (Tsai et al., 2025). AnoLLM serializes each row into a text template, discretizes numerical values into bins, and fine-tunes a LLM on normal data via next-token prediction. At inference time, it scores anomalies using negative log-likelihood, together with procedures to better handle mixed-type tables. Despite this progress, AnoLLM can be viewed as a one-shot SFT-style adaptation of a gener-

---

[1]Hanoi University of Science and Technology, Hanoi, Vietnam [2]Applied Artificial Intelligence Initiative, Deakin University, Australia [3]Monash University, Clayton, VIC 3800, Australia. Correspondence to: Linh Ngo Van <linhnv@soict.hust.edu.vn>, Trung Le <trunglm@monash.edu>.

*Proceedings of the 43$^{rd}$ International Conference on Machine Learning*, Seoul, South Korea. PMLR 306, 2026. Copyright 2026 by the author(s).

ative model to normal data, leaving a meaningful gap for unsupervised anomaly detection. Maximizing likelihood on normal training data provides only a weak signal for tightening the boundary of normality: once model fits the dominant regularities of the training corpus, further gains in separating subtle, near-normal anomalies are often limited and performance can plateau. These limitations motivate an iterative refinement mechanism that more directly reduces the mismatch between model-induced distribution and the true normal-data distribution, while remaining compatible with likelihood-based detection.

Self-play provides a principled route to such iterative refinement. Recent advances such as SPIN (Chen et al., 2024) improve an SFT-initialized model by repeatedly contrasting real data with the model's own generations: at each iteration, the current model produces synthetic samples, and the next update is driven by a discriminator-like objective that favors genuine data over model-generated counterparts, providing a learning signal without additional annotations. In unsupervised tabular AD, this provides an iterative refinement signal: a critic distinguishes real normal samples from model-generated ones, and the policy is updated so that future generations better match the normal-data distribution. Unlike SPIN, which optimizes a single fixed IPM-like divergence, we extend this idea to a broader family of *f*-divergences. Different divergences induce different distribution-matching behavior—for example, KL is more mode-covering, whereas reverse KL is more mode-seeking—which is useful in anomaly detection because the geometry of the normal data manifold can vary substantially across domains. Repeating this loop can improve the quality and calibration of likelihood-based anomaly scores without requiring anomaly labels.

In this paper, we introduce **DiSPaT**, a **Di**vergence-driven **S**elf-**Pl**ay framework for unsupervised **T**abular anomaly detection with LLMs. **DiSPaT** treats each mixed-type row as a serialized token sequence and iteratively refines the model's understanding of normality through an adversarial self-play game. Formally, we optimize the LLM to minimize the *f*-divergence between the empirical distribution of normal data and the model-induced distribution. Conceptually, this is achieved via an alternating optimization loop: in the *critic step*, a critic contrasts real normal sequences with samples from the current policy; in the *policy step*, the LLM is updated using this contrastive signal so that its induced distribution moves closer to the normal-data distribution. In the final implementable objective, the critic is absorbed into an implicit log-ratio parameterization, yielding a single optimization step per iteration. Through this iterative process, the model progressively aligns its induced distribution with the true normal-data distribution. Finally, anomalies are identified based on the likelihood scores of the refined model. Our contributions are summarized as follows:

- We propose a principled theoretical framework for unsupervised tabular anomaly detection via LLMs by reformulating the training objective as minimizing *f*-divergence between normal data distribution and model's generations. This formulation transcends the limitations of one-shot SFT, providing a mechanism that explicitly targets the distributional mismatch between model and true data manifold.

- We introduce **DiSPaT**, a practical algorithm that implements this framework via a self-play loop with an implicit critic. It preserves the critic–policy interpretation while reducing each iteration to a single tractable optimization step, enabling the model to capture complex tabular dependencies without anomalous supervision.

- Extensive evaluations on mixed-type benchmarks demonstrate that the proposed approach outperforms state-of-the-art baselines, including recent LLM-based and specialized deep anomaly detectors. Furthermore, detailed ablation studies on iteration depth and divergence choices validate the efficacy of the self-play mechanism.

## 2. Background

### 2.1. Problem Formulation

We study uncontaminated, unsupervised anomaly detection on tabular data. Let $X \in \mathbb{R}^{n \times d}$ denote a training data matrix consisting solely of normal samples, where each row $\mathbf{x}_i \in \mathcal{X}$ for $i \in \{1, \dots, n\}$ is a sample with $d$ features, and $x_{i,j}$ for $j \in \{1, \dots, d\}$ represents attributes of different types. The objective is to train an anomaly detector $f : \mathcal{X} \to \mathbb{R}$ using only $\mathcal{X}$. The detector assigns an anomaly score to each input, where higher values indicate a greater likelihood of being an anomaly. At evaluation time, a labeled test set $D' = \{(x'_m, y'_m)\}_{m=1}^{M}$ with $y'_m \in \{0, 1\}$ is provided, and compute $f(x'_m)$ for each test sample. A threshold can be employed to classify whether a sample is an anomaly based on these scores.

In LLM-based anomaly detection frameworks like AnoLLM (Tsai et al., 2025), each tabular row $x_i$ is represented as a serialized token sequence. Specifically, each feature $x_{i,j}$ is transformed into a natural language phrase "$c_j$ is $E(x_{i,j})$", where $c_j$ is column name and $E(\cdot)$ is an encoder for pre-processing values. After handling missing values and column names, a permutation $\pi$ is dynamically sampled from $S_d$ during training, where $S_d$ denotes the symmetric group on $d$ elements, i.e., the group of all permutations of the set $\{1, 2, \dots, d\}$ under composition[1]. The resulting

---

[1]For example, with $d = 3$, the symmetric group $S_3$ contains $3! = 6$ permutations: the identity $(1)(2)(3)$, transpositions $(1, 2)$, $(1, 3)$, $(2, 3)$, and 3-cycles $(1, 2, 3)$, $(1, 3, 2)$. In our notation, $\pi(i)$ denotes the image of element $i$ under permutation $\pi$.

serialized sequence is defined as:

$$\mathbf{S}(\pi, \mathbf{x}_i) = \text{``}c_{\pi(1)} \text{ is } E(x_{i,\pi(1)}), \ldots, c_{\pi(d)} \text{ is } E(x_{i,\pi(d)})\text{''}$$

The resulting set of serialized tabular strings is denoted as $\mathcal{S} = \{\mathbf{S}(\pi, \mathbf{x}_i) \mid \pi \in S_d, i \in \{1, \ldots, n\}\}$. For each string $\mathbf{s} \in \mathcal{S}$, data is transformed into a token sequence $(w_0, w_1, \ldots, w_{l(\mathbf{s})}, w_{l(\mathbf{s})+1})$, where $w_0$ and $w_{l(\mathbf{s})+1}$ represent the beginning-of-sequence (BoS) and end-of-sequence (EoS) tokens, respectively. The fine-tuning process minimizes causal language modeling loss:

$$\mathcal{L}(\boldsymbol{\theta}) = \mathbb{E}_{\mathbf{s} \in \mathcal{S}} \left[ -\sum_{k=1}^{l(\mathbf{s})+1} \log \pi_{\boldsymbol{\theta}}(w_k \mid w_{0:k-1}) \right] \quad (1)$$

where $\boldsymbol{\theta}$ represents the parameters of the model. Through this objective, LLM learns to predict subsequent column values based on previously observed features, leveraging its inherent semantic knowledge to capture the underlying structure of normal data. In the inference stage, anomaly score for a test sample $\mathbf{x}'$ is computed by averaging negative log-likelihood across $r$ different permutations:

$$f_{\boldsymbol{\theta}}(\mathbf{x}') = -\frac{1}{r} \sum_{j=1}^{r} \sum_{k=1}^{l(\mathbf{S}(\pi_j, \mathbf{x}'))+1} \log \pi_{\boldsymbol{\theta}}(w_k^{(j)} \mid w_{0:k-1}^{(j)}) \quad (2)$$

where $l(\mathbf{S}(\pi_j, \mathbf{x}'))$ denotes number of tokens used to encode tabular features under $j$-th permutation. Here, $\pi_1, \ldots, \pi_r \sim S_d$ denote distinct permutations of $\{1, 2, \ldots, d\}$, applied consistently across all samples.

### 2.2. *f*-Divergence

To quantify the discrepancy between two probability distributions, we utilize the family of statistical measures known as $f$-divergences (Csiszár & Shields, 2004; Liese & Vajda, 2006). Given two distributions $P$ and $Q$ possessing, respectively, absolutely continuous density functions $p$ and $q$ with respect to a base measure defined on the domain $\mathcal{X}$, the $f$-divergence is defined as:

$$D_f(P \| Q) = \int_{\mathcal{X}} q(x) f\left(\frac{p(x)}{q(x)}\right) dx, \quad (3)$$

where $f : \mathbb{R}^+ \to \mathbb{R}$ is a convex, lower-semi continuous function satisfying $f(1) = 0$.

### 2.3. Self-Play Fine-Tuning

Self-Play Fine-Tuning (SPIN) (Chen et al., 2024; Le et al., 2026; Wang et al., 2026b) is a framework designed to enhance language model performance by employing a two-player game mechanism. At iteration $t + 1$, model from the previous iteration, $\pi_{\boldsymbol{\theta}_t}$, acts as the opponent to generate

synthetic responses $\mathbf{y}'$ for a given input $\mathbf{x}$. Current model $\pi_{\boldsymbol{\theta}}$ (the main player) is trained to distinguish these self-generated responses from ground-truth responses $\mathbf{y}$ sampled from real data distribution $\mathbb{P}_d$. The training objective maximizes the probabilistic gap between real and synthetic data distributions by minimizing the expected logistic loss $\ell(t) = \log(1 + \exp(-t))$:

$$\mathcal{L}_{\text{SPIN}}(\boldsymbol{\theta}) = \mathbb{E}\left[\ell\left(\lambda \log \frac{\pi_{\boldsymbol{\theta}}(\mathbf{y} \mid \mathbf{x})}{\pi_{\boldsymbol{\theta}_t}(\mathbf{y} \mid \mathbf{x})} - \lambda \log \frac{\pi_{\boldsymbol{\theta}}(\mathbf{y}' \mid \mathbf{x})}{\pi_{\boldsymbol{\theta}_t}(\mathbf{y}' \mid \mathbf{x})}\right)\right] \quad (4)$$

where the expectation is taken over $\mathbf{x}, \mathbf{y} \sim \mathbb{P}_d$ and $\mathbf{y}' \sim \pi_{\boldsymbol{\theta}_t}$, and $\lambda > 0$ serves as a regularization parameter. Theoretically, the global optimum of this objective is attained if and only if the model's policy $\pi_{\boldsymbol{\theta}}$ fully aligns with the target distribution $\mathbb{P}_d$, thereby facilitating iterative self-improvement without requiring additional human annotations.

## 3. Methodology

In LLM-based anomaly detection, AnoLLM (Tsai et al., 2025) fine-tunes an LLM to maximize likelihood of serialized normal samples, learning a generative description of normality. However, this one-shot fine-tuning operates solely on normal instances without abnormal examples. Consequently, the learned decision boundary tends to be loose, lacking discriminative signal to tightly delineate normality. Prior work has shown that incorporating abnormal samples or limited label information during training yields a tighter description of normal data (Le et al., 2010; 2013; Ruff et al., 2020; Pang et al., 2019; Ding et al., 2022). In settings where true anomalies are unavailable, we propose a self-play mechanism that synthetically generates pseudo-abnormal examples to address this limitation. Specifically, at each iteration, model-generated samples $\boldsymbol{y}'$ from earlier policies deviate from real normal continuations $\boldsymbol{y}$ and therefore serve as pseudo-abnormal, or more precisely, *not-normal*, references. A critic contrasts these samples with real normal data, providing a discriminative signal that sharpens the model's description of normality without requiring actual anomaly labels. As self-play progresses and $\boldsymbol{y}'$ moves closer to $\boldsymbol{y}$, the model is forced to make increasingly fine-grained distinctions near the normal data manifold, leading to a progressively tighter boundary of normality. Importantly, we do not assume that these synthetic samples approximate the unknown anomaly distribution. Their role is not to model anomalies, but to provide progressively harder not-normal references that tighten the normality boundary. This viewpoint is consistent with one-class learning: OCSVM (Schölkopf et al., 2001) separates normal data from a single origin in feature space, even though that origin does not resemble any real anomaly. DiSPaT follows

the same principle, but replaces a fixed origin with model-generated non-normal samples that evolve across iterations, yielding a richer signal for boundary refinement.

Let $\mathbb{P}_d$ denote the distribution over ground-truth normal sequences $(\boldsymbol{x}, \boldsymbol{y})$, where $\boldsymbol{x}$ is a context and $\boldsymbol{y} \sim \mathbb{P}_d(\cdot \mid \boldsymbol{x})$ is real continuation. Correspondingly, $\mathbb{P}_{\boldsymbol{\theta}}$ represents model-induced distribution with synthetic continuations $\boldsymbol{y}' \sim \pi_{\boldsymbol{\theta}}(\cdot \mid \boldsymbol{x})$. Our goal is to learn $\mathbb{P}_{\boldsymbol{\theta}}$ that closely aligns with $\mathbb{P}_d$ through iterative self-play. At each iteration, one can view a critic $T$ as distinguishing real sequences $\boldsymbol{y}$ from model-generated sequences $\boldsymbol{y}' \sim \pi_{\boldsymbol{\theta}_k}$. The policy $\pi_{\boldsymbol{\theta}}$ is then updated to maximize scores under this critic, thereby pushing $\boldsymbol{y}'$ toward $\boldsymbol{y}$ and the distribution $\mathbb{P}_{\boldsymbol{\theta}}$ toward $\mathbb{P}_d$. As we show below, this critic admits an implicit log-ratio parameterization, so the conceptual critic step and the policy step can be unified into a single optimization. This adversarial mechanism provides the discriminative learning signal absent in likelihood-only training, enabling the model to learn a tighter description of normality. This motivates our objective of minimizing the $f$-divergence:

$$\min_{\boldsymbol{\theta}} \; D_f\left(\mathbb{P}_d, \mathbb{P}_{\boldsymbol{\theta}}\right) \tag{5}$$

Minimizing this divergence aims to align the model's synthetic data distribution with the true data distribution. In what follows, we present a theory to tightly approximate $D_f(\mathbb{P}_d, \mathbb{P}_{\boldsymbol{\theta}})$ in Theorem 3.1.

**Theorem 3.1.** *Given a lower semi-continuous convex function $f$ that is differentiable and strictly convex on the interior of its domain with Fenchel conjugate $f^*$, and assuming $\mathbb{P}_d \ll \mathbb{P}_{\boldsymbol{\theta}}$ with $T^*$ and $f^* \circ T^*$ bounded on the support, then for any function family $\mathcal{T}$ we have the following inequalities:*

*(i)* $\displaystyle D_f(\mathbb{P}_d, \mathbb{P}_{\boldsymbol{\theta}}) \geq \max_{T \in \mathcal{T}} \Big\{ \mathbb{E}_{\mathbb{P}_d}[T(\boldsymbol{x}, \boldsymbol{y})]$
$$- \mathbb{E}_{\mathbb{P}_{\boldsymbol{\theta}}}[f^*(T(\boldsymbol{x}, \boldsymbol{y}'))] \Big\}$$

*(ii)* $\displaystyle D_f(\mathbb{P}_d, \mathbb{P}_{\boldsymbol{\theta}}) \leq \max_{T \in \mathcal{T}} \Big\{ \mathbb{E}_{\mathbb{P}_d}[T(\boldsymbol{x}, \boldsymbol{y})]$
$$- \mathbb{E}_{\mathbb{P}_{\boldsymbol{\theta}}}[f^*(T(\boldsymbol{x}, \boldsymbol{y}'))] \Big\} + d(T^*, \mathcal{T})$$

*where $T^*(\boldsymbol{x}, \boldsymbol{y}) = \nabla f\left(\frac{p_d(\boldsymbol{x}, \boldsymbol{y})}{p_{\boldsymbol{\theta}}(\boldsymbol{x}, \boldsymbol{y})}\right)$ and the distance $d(T^*, \mathcal{T})$ between $T^*$ and family $\mathcal{T}$ is defined as:*

$$d(T^*, \mathcal{T}) := \min_{T \in \mathcal{T}} \left\{ \|T - T^*\|_\infty + \|f^* \circ T - f^* \circ T^*\|_\infty \right\}$$

*in which we define $\|G\|_\infty = \max_{(\boldsymbol{x}, \boldsymbol{y})} |G(\boldsymbol{x}, \boldsymbol{y})|$.*

From Theorem 3.1 (proof in Appendix A), the gap between $D_f(\mathbb{P}_d, \mathbb{P}_{\boldsymbol{\theta}})$ and the variational term $\max_{T \in \mathcal{T}} \{\mathbb{E}_{\mathbb{P}_d}[T(\boldsymbol{x}, \boldsymbol{y})] - \mathbb{E}_{\mathbb{P}_{\boldsymbol{\theta}}}[f^*(T(\boldsymbol{x}, \boldsymbol{y}'))]\}$ is controlled by $d(T^*, \mathcal{T})$. Since $\mathcal{T}$ is parameterized by an

expressive LLM, $d(T^*, \mathcal{T})$ is small and the variational term closely matches $D_f(\mathbb{P}_d, \mathbb{P}_{\boldsymbol{\theta}})$. We therefore use it as a tractable proxy:

$$D_f(\mathbb{P}_d, \mathbb{P}_{\boldsymbol{\theta}}) \approx \max_{T \in \mathcal{T}} \Big\{ \mathbb{E}_{\mathbb{P}_d}\big[T(\boldsymbol{x}, \boldsymbol{y})\big] \\ - \mathbb{E}_{\mathbb{P}_{\boldsymbol{\theta}}}\big[f^*(T(\boldsymbol{x}, \boldsymbol{y}'))\big] \Big\} \tag{6}$$

Going back to our main problem, the main OP in (5) can be rewritten as:

$$\min_{\boldsymbol{\theta}} \max_{T \in \mathcal{T}} \left\{ \mathbb{E}_{\mathbb{P}_d}[T(\boldsymbol{x}, \boldsymbol{y})] - \mathbb{E}_{\mathbb{P}_{\boldsymbol{\theta}}}[f^*(T(\boldsymbol{x}, \boldsymbol{y}'))] \right\} \tag{7}$$

To further develop our theoretical framework, we consider a specific family of convex functions $f$ whose Fenchel conjugate $f^*$ is non-decreasing. This choice lets the monotone $f^*$ preserve the maximizer when passing from $f^*(T_k(\boldsymbol{x}, \boldsymbol{y}'))$ to $T_k(\boldsymbol{x}, \boldsymbol{y}')$ in the policy step, leading to more tractable formulations as detailed in the following derivations.

**Theorem 3.2.** *Consider convex functions of the form:*

$$f(u) = \begin{cases} g(u), & u \geq 0, \\ +\infty, & u < 0, \end{cases} \tag{8}$$

*where $g : [0, +\infty) \to \mathbb{R} \cup \{+\infty\}$ is proper and lower semi-continuous, and differentiable and strictly convex on $(0, +\infty)$. Let $g'_+(0) = \lim_{u \to 0^+} g'(u)$ and $g'_\infty = \lim_{u \to +\infty} g'(u)$. For $g'_+(0) < v < g'_\infty$, define $g^*(v) = u^* v - g(u^*)$, where $u^* > 0$ is the unique solution of $g'(u^*) = v$. Then the Fenchel conjugate of $f$ is given by:*

$$f^*(v) = \begin{cases} -g(0), & v \leq g'_+(0), \\ g^*(v), & g'_+(0) < v < g'_\infty, \\ +\infty, & v > g'_\infty, \end{cases} \tag{9}$$

*where $\partial g(0)$ denotes the subdifferential of $g$ at 0, so $g'_+(0) = \sup \partial g(0)$ (with the convention $\sup \varnothing = -\infty$). The boundary case $v = g'_\infty$ is discussed in Appendix A. In particular, $f^*$ is non-decreasing.*

In the remainder of this section, we consider only convex functions $f$ of the form defined in Theorem 3.2 (proof in Appendix A), ensuring $f^*$ is non-decreasing. To handle min-max problem in (7), similar to GAN and $f$-GAN, we conceptually alternate between critic $T$ and LLM policy $\pi_{\boldsymbol{\theta}}$ as follows:

**(1) Critic Update.** Assume that at the current iteration $k$, we have current LLM model $\pi_{\boldsymbol{\theta}_k}$, and we update critic $T_k$ to maximize:

$$\mathbb{E}_{\mathbb{P}_d}[T(\boldsymbol{x}, \boldsymbol{y})] - \mathbb{E}_{\mathbb{P}_{\boldsymbol{\theta}_k}}[f^*(T(\boldsymbol{x}, \boldsymbol{y}'))] \tag{10}$$

In other words, we learn the critic function to optimize:

$$\max_{T \in \mathcal{T}} \left\{ \mathbb{E}_{\mathbb{P}_d} \left[ T(\boldsymbol{x}, \boldsymbol{y}) \right] - \mathbb{E}_{\mathbb{P}_{\boldsymbol{\theta}_k}} \left[ f^*(T(\boldsymbol{x}, \boldsymbol{y}')) \right] \right\}$$
$$= \max_{T \in \mathcal{T}} \left\{ \mathbb{E}_{\boldsymbol{x}} \left[ T(\boldsymbol{x}, \boldsymbol{y}) - f^*(T(\boldsymbol{x}, \boldsymbol{y}')) \right] \right\} \quad (11)$$

where $\boldsymbol{y} \sim \mathbb{P}_d(\cdot \mid \boldsymbol{x})$ and $\boldsymbol{y}' \sim \pi_{\boldsymbol{\theta}_k}(\cdot \mid \boldsymbol{x})$. Instead of maximizing (11), we propose to minimize:

$$\min_{T \in \mathcal{T}} \left\{ \mathbb{E}_{\boldsymbol{x}} \left[ \ell \left( T(\boldsymbol{x}, \boldsymbol{y}) - f^*(T(\boldsymbol{x}, \boldsymbol{y}')) \right) \right] \right\}, \quad (12)$$

where $\ell$ is a non-linear and decreasing function (e.g., $\ell(t) = \log(1 + e^{-t})$). First, this stabilizes the critic loss, as logistic loss is smooth and non-negative, preventing excessive growth in critic values during training. In addition, the resulting objective resembles those used in preference-optimization methods (Rafailov et al., 2023; Chen et al., 2024), which have proven effective and offer tractability with stable gradient properties.

**(2) Policy Update.** Subsequently, we update LLM model $\pi_{\boldsymbol{\theta}}$ to move $\mathbb{P}_{\boldsymbol{\theta}}$ closer to $\mathbb{P}_d$ by performing the outer minimization, leading to:

$$\max_{\boldsymbol{\theta}} \mathbb{E}_{\mathbb{P}_{\boldsymbol{\theta}}} \left[ f^*(T_k(\boldsymbol{x}, \boldsymbol{y}')) \right] \quad (13)$$

This updates the model to maximize $f^*(T_k(\boldsymbol{x}, \boldsymbol{y}'))$ for $\boldsymbol{y}' \sim \pi_{\boldsymbol{\theta}}(\cdot \mid \boldsymbol{x})$, thereby pushing synthetic samples $(\boldsymbol{x}, \boldsymbol{y}')$ into high-score regions defined by the critic, effectively making $\mathbb{P}_{\boldsymbol{\theta}}$ converge toward $\mathbb{P}_d$. Because $f^*$ is non-decreasing, raising the critic score $T_k(\boldsymbol{x}, \boldsymbol{y}')$ also raises $f^*(T_k(\boldsymbol{x}, \boldsymbol{y}'))$. This motivates replacing (13) with the KL-regularized surrogate below, whose optimum has a closed form:

$$\max_{\pi} \left\{ \mathbb{E}_{\boldsymbol{x}, \boldsymbol{y}' \sim \pi(\cdot \mid \boldsymbol{x})} \left[ T_k(\boldsymbol{x}, \boldsymbol{y}') \right] \right.$$
$$\left. - \beta \mathbb{E}_{\boldsymbol{x}} \left[ D_{KL}(\pi(\cdot \mid \boldsymbol{x}) \| \pi_{\boldsymbol{\theta}_k}(\cdot \mid \boldsymbol{x})) \right] \right\} \quad (14)$$

whose optimal solution is:

$$\pi^*(\boldsymbol{y}' \mid \boldsymbol{x}) = \frac{1}{Z(\boldsymbol{x})} \pi_{\boldsymbol{\theta}_k}(\boldsymbol{y}' \mid \boldsymbol{x}) \exp\left\{ \beta^{-1} T_k(\boldsymbol{x}, \boldsymbol{y}') \right\} \quad (15)$$

where $Z(\boldsymbol{x})$ is the partition function. From (15), it follows that for each policy $\pi_{\boldsymbol{\theta}}$, there exists $T_k(\boldsymbol{x}, \boldsymbol{y}' \mid \boldsymbol{\theta}) = \beta \log \frac{\pi_{\boldsymbol{\theta}}(\boldsymbol{y}' \mid \boldsymbol{x})}{\pi_{\boldsymbol{\theta}_k}(\boldsymbol{y}' \mid \boldsymbol{x})} + \beta \log Z(\boldsymbol{x})$ such that $T_k(\boldsymbol{x}, \boldsymbol{y}' \mid \boldsymbol{\theta})$ and $\pi_{\boldsymbol{\theta}}$ form the solution pair of the OP in (14). We have the following proposition.

**Proposition 3.3.** *Consider the following OP, obtained by substituting the implicit-critic parameterization induced by*

*(15) into the critic objective (12):*

$$\min_{\boldsymbol{\theta}} \mathbb{E}_{\boldsymbol{x}} \left[ \ell \left( \beta \log \frac{\pi_{\boldsymbol{\theta}}(\boldsymbol{y} \mid \boldsymbol{x})}{\pi_{\boldsymbol{\theta}_k}(\boldsymbol{y} \mid \boldsymbol{x})} + \beta \log Z(\boldsymbol{x}) \right. \right.$$
$$\left. \left. - f^* \left( \beta \log \frac{\pi_{\boldsymbol{\theta}}(\boldsymbol{y}' \mid \boldsymbol{x})}{\pi_{\boldsymbol{\theta}_k}(\boldsymbol{y}' \mid \boldsymbol{x})} + \beta \log Z(\boldsymbol{x}) \right) \right) \right] \quad (16)$$

The partition function $Z(\boldsymbol{x})$ sums over the entire space of generated sequences and is therefore intractable to compute exactly. We thus replace (16) with the surrogate in (17), which omits $\beta \log Z(\boldsymbol{x})$ from *both* chosen and rejected scores:

$$\min_{\boldsymbol{\theta}} \mathbb{E}_{\boldsymbol{x}} \left[ \ell \left( \beta \log \frac{\pi_{\boldsymbol{\theta}}(\boldsymbol{y} \mid \boldsymbol{x})}{\pi_{\boldsymbol{\theta}_k}(\boldsymbol{y} \mid \boldsymbol{x})} \right. \right.$$
$$\left. \left. - f^* \left( \beta \log \frac{\pi_{\boldsymbol{\theta}}(\boldsymbol{y}' \mid \boldsymbol{x})}{\pi_{\boldsymbol{\theta}_k}(\boldsymbol{y}' \mid \boldsymbol{x})} \right) \right) \right] \quad (17)$$

This objective encourages the model to increase the likelihood of real normal samples $\boldsymbol{y}$ while maintaining or improving upon the reference model's generation of $\boldsymbol{y}'$, progressively refining the model's understanding of the normal data distribution.

**Implicit Critic Implementation.** Although the derivation is presented as a critic step followed by a policy step, the final training algorithm does not require a separate inner-loop optimization for $T$. Following the same principle as DPO (Rafailov et al., 2023) and SPIN (Chen et al., 2024), the KL-regularized policy problem in (14) yields the closed-form solution in (15), which implies:

$$T_k(\boldsymbol{x}, \boldsymbol{y}' \mid \boldsymbol{\theta}) = \beta \log \frac{\pi_{\boldsymbol{\theta}}(\boldsymbol{y}' \mid \boldsymbol{x})}{\pi_{\boldsymbol{\theta}_k}(\boldsymbol{y}' \mid \boldsymbol{x})} + \beta \log Z(\boldsymbol{x}).$$

Hence the critic is an implicit function of the policy parameters. Substituting this parameterization back into the critic objective (12) yields (16), and omitting the partition term $\beta \log Z(\boldsymbol{x})$ gives the implementable surrogate (17). A single optimization over $\boldsymbol{\theta}$ thus realizes both the conceptual critic and policy updates, and Algorithm 1 accordingly performs one parameter update per iteration.

In practice, the model at iteration $k$ may occasionally generate synthetic samples $\boldsymbol{y}'$ that closely resemble true normal data. When this occurs, the probability ratio $\frac{\pi_{\boldsymbol{\theta}}(\boldsymbol{y}' \mid \boldsymbol{x})}{\pi_{\boldsymbol{\theta}_k}(\boldsymbol{y}' \mid \boldsymbol{x})}$ approaches 1, indicating that both the current and reference policies assign similar likelihoods to $\boldsymbol{y}'$. Penalizing such samples as pseudo-anomalies would be counterproductive, as they already lie close to the normal data manifold. To address this, we apply a clamping operation:

$$\tilde{\rho}^l = \beta \log \min \left\{ \frac{\pi_{\boldsymbol{\theta}}(\boldsymbol{y}' \mid \boldsymbol{x})}{\pi_{\boldsymbol{\theta}_k}(\boldsymbol{y}' \mid \boldsymbol{x})}, 1 - \epsilon \right\} \quad (18)$$

**Algorithm 1** DiSPaT

**Input:** Tabular dataset $X = \{\mathbf{x}_i\}_{i=1}^n$, SFT-initialized LLM $\pi_{\boldsymbol{\theta}_0}$, iterations $T$, non-decreasing Fenchel conjugate $f^*$, regularization $\beta$, clamping threshold $\epsilon$

1: **for** $i = 1, \ldots, n$ **do**
2:     Sample permutation $\pi \sim S_d$
3:     Serialize row $\mathbf{x}_i$ into context-continuation pair $(\mathbf{x}_i, \mathbf{y}_i) = \mathbf{S}(\pi, \mathbf{x}_i)$
4: **end for**
5: **for** $t = 0, \ldots, T - 1$ **do**
6:     *// Generate synthetic samples to serve as pseudo-abnormal*
7:     **for** $i = 1, \ldots, n$ **do**
8:         Sample synthetic continuation $\mathbf{y}_i' \sim \pi_{\boldsymbol{\theta}_t}(\cdot \mid \mathbf{x}_i)$
9:     **end for**
10:    *// Single implicit-critic update to minimize f-divergence*
11:    Compute chosen reward: $\rho_i^w \leftarrow \beta \log \frac{\pi_{\boldsymbol{\theta}}(\mathbf{y}_i|\mathbf{x}_i)}{\pi_{\boldsymbol{\theta}_t}(\mathbf{y}_i|\mathbf{x}_i)}$
12:    Compute rejected reward with clamping:
$$\tilde{\rho}_i^l \leftarrow \beta \log \min \left\{ \frac{\pi_{\boldsymbol{\theta}}(\mathbf{y}_i' \mid \mathbf{x}_i)}{\pi_{\boldsymbol{\theta}_t}(\mathbf{y}_i' \mid \mathbf{x}_i)}, 1 - \epsilon \right\}$$
13:    Update: $\boldsymbol{\theta}_{t+1} \leftarrow \arg\min_{\boldsymbol{\theta}} \sum_{i=1}^n \ell\left(\rho_i^w - f^*(\tilde{\rho}_i^l)\right)$
14: **end for**
**Output:** Trained model $\boldsymbol{\theta}_T$

where $\epsilon > 0$ is a small threshold. This clamping effectively treats synthetic samples that are already similar to normal data as normal, avoiding over-penalization of such "good attempts" and ensuring stable optimization. The algorithm is presented in Algorithm 1. **Examples of generated samples across iterations are shown in Appendix B.8.**

## 4. Experiments

**Datasets.** Standard anomaly detection benchmarks such as ADBench (Han et al., 2022) and the ODDS library (Rayana, 2016) are dominated by purely numerical features. Following AnoLLM (Tsai et al., 2025), that include textual, numerical, or categorical features, drawn from the ODDS library (Rayana, 2016), a fraud detection benchmark (Grover et al., 2022), and Kaggle. To further validate the robustness of DiSPaT, we additionally evaluate on 30 numerical datasets from ODDS library. Dataset statistics for all datasets are summarized in Appendix B.7, and detailed ODDS results are reported in Appendix B.4.

**Baselines.** We compare DiSPaT against 12 established tabular anomaly detection methods. Classical approaches rely on techniques such as tree-based isolation, linear projection, nearest-neighbor distances, and statistical distribution modeling; these include IForest (Liu et al., 2008), PCA (Shyu et al., 2003), $k$NN (Ramaswamy et al., 2000), and ECOD (Li et al., 2022). Deep learning methods lever-

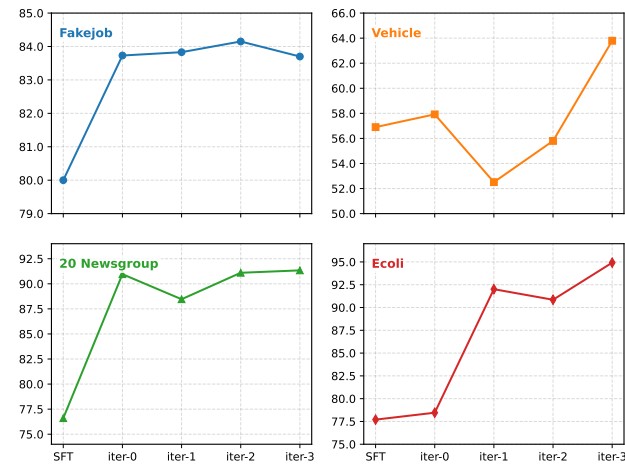

*Figure 1.* AUC-ROC scores (%) of DiSPaT across self-play iterations on four datasets using SmolLM-135M. SFT indicates initial model, and iter-$k$ refers to model after $k$ iterations of training.

age neural architectures, reconstruction-based learning, self-supervised objectives with data augmentation or contrastive learning, and generative diffusion processes; these include DeepSVDD (Ruff et al., 2018), RCA (Liu et al., 2021), SLAD (Xu et al., 2023), GOAD (Bergman & Hoshen, 2020), NeuTral (Qiu et al., 2021), ICL (Shenkar & Wolf, 2022), DTE (Livernoche et al., 2024), and REPEN (Pang et al., 2018). We also include the LLM-based method AnoLLM (Tsai et al., 2025) as a baseline.

**Implementation and Evaluation.** We employ SmolLM-135M, SmolLM-360M, and SmolLM-1.7B as backbone LLMs, which are among the strongest publicly available open-weight small-scale models. Following AnoLLM (Tsai et al., 2025), we fine-tune all models for 2000 steps (20000 steps for the Fakejob dataset) and select learning rates from $\{1 \times 10^{-3}, 5 \times 10^{-5}\}$ per dataset. We tune coefficient $\beta$ and clamping threshold $\epsilon$ for each dataset via grid search. For evaluation, we follow established benchmarks (Shenkar & Wolf, 2022; Xu et al., 2023): 50% of normal data for training, with the remaining normal instances and all anomalies forming the test set. We report AUC-ROC, F1-score, and AUC-PR. Anomaly scores are computed as negative log-likelihood of serialized token sequences, averaged over $r = 21$ random column permutations (Tsai et al., 2025). Detailed hyperparameter settings and implementation specifics are provided in Appendix B.1, and a detailed analysis of training and inference cost is given in Appendix B.2.

### 4.1. Main Results

In all experiments, we adopt KL divergence with $f^*(v) = e^v - 1$, as it yields the best overall performance among the $f$-divergence variants; detailed comparisons are provided in Section 4.2. Tables 1, 2 and 3 present the performance

*Table 1.* Detailed AUC-ROC scores (%) comparing DiSPaT against baselines on the six-dataset benchmark. Best and second-best results per dataset are highlighted in **bold** and underlined, respectively. Values marked with ↑ and ↓ indicate where DiSPaT improves or degrades compared to AnoLLM with the same model size. All baseline results are reported directly from (Tsai et al., 2025).

| Methods\Datasets | Fakejob | Lymphography | Seismic | Vehicle | 20 Newsgroups | Ecoli | Average |
|---|---|---|---|---|---|---|---|
| *Classical methods* | | | | | | | |
| Iforest | 75.5% | 67.3% | 69.2% | 49.6% | 62.3% | 85.6% | 68.3% |
| PCA | 72.4% | 82.6% | 69.2% | 50.9% | 62.3% | 85.6% | 70.5% |
| KNN | 63.6% | 86.0% | 73.8% | 52.4% | 60.5% | 87.7% | 70.7% |
| ECOD | 51.2% | 83.0% | 69.2% | 50.9% | 62.0% | 77.6% | 65.7% |
| *Deep Learning based methods* | | | | | | | |
| DeepSVDD | 56.1% | 89.9% | 71.3% | 50.5% | 59.7% | 88.7% | 69.4% |
| RCA | 62.9% | 91.9% | 72.7% | 53.1% | 54.6% | 88.3% | 70.6% |
| SLAD | 60.3% | 96.4% | 71.4% | 55.6% | 64.0% | 88.2% | 72.7% |
| GOAD | 56.6% | 81.7% | 71.7% | 51.2% | 63.0% | 88.1% | 68.7% |
| NeuTral | 54.8% | 84.7% | 68.1% | 50.7% | 65.8% | 86.0% | 68.4% |
| ICL | 69.9% | 82.7% | 71.9% | 50.1% | 67.1% | 88.7% | 71.7% |
| DTE | 54.8% | 90.9% | 71.4% | 51.2% | 60.0% | 82.1% | 68.4% |
| REPEN | 65.3% | 80.8% | 72.4% | 51.3% | 57.4% | 87.0% | 69.0% |
| *LLM-based methods* | | | | | | | |
| AnoLLM (SmolLM-135M) | 80.0% | 96.8% | 71.2% | 56.9% | 76.6% | 77.7% | 76.5% |
| DiSPaT | 84.2%$^\uparrow$ | **100.0%**$^\uparrow$ | 73.2%$^\uparrow$ | **63.8%**$^\uparrow$ | **91.9%**$^\uparrow$ | 94.9%$^\uparrow$ | 84.6%$^\uparrow$ |
| AnoLLM (SmolLM-360M) | 81.4% | 99.5% | 74.6% | 55.5% | 75.2% | 80.4% | 77.8% |
| DiSPaT | **88.3%**$^\uparrow$ | **100.0%**$^\uparrow$ | **75.7%**$^\uparrow$ | **64.2%**$^\uparrow$ | 87.6%$^\uparrow$ | **96.8%**$^\uparrow$ | **85.4%**$^\uparrow$ |
| AnoLLM (SmolLM-1.7B) | 80.2% | 99.5% | 74.0% | 56.0% | 77.4% | 79.1% | 77.7% |
| DiSPaT | 85.2%$^\uparrow$ | 99.8%$^\uparrow$ | 74.6%$^\uparrow$ | 59.0%$^\uparrow$ | 87.1%$^\uparrow$ | 95.1%$^\uparrow$ | 83.5%$^\uparrow$ |

comparison on the six-dataset benchmark. **DiSPaT** consistently delivers the best results across all datasets and model scales. Specifically, DiSPaT improves over AnoLLM by **8.1%**, **7.6%**, and **5.8%** in average AUC-ROC for the 135M, 360M, and 1.7B backbones, respectively. Similar improvements are observed on AUC-PR and F1 metrics: DiSPaT achieves **13.9%**, **14.1%**, and **7.9%** improvements in AUC-PR, and **13.9%**, **15.6%**, and **7.9%** improvements in F1 for three model sizes. Compared to classical and deep learning baselines, DiSPaT substantially outperforms the best-performing baseline across all metrics: at least **12.7%** higher in AUC-ROC, **12.1%** in AUC-PR, and **15.4%** in F1. Additional comparisons with recent tabular anomaly detection baselines are reported in Appendix B.3. This highlights the advantage of LLM-based approaches for tabular data that may include textual, numerical, or categorical attributes, where traditional methods require extensive feature engineering to jointly model different attribute types.

Figure 1 illustrates the effect of iterative self-play training. Starting from the SFT-initialized model, performance generally improves as iterations increase, with most datasets showing gains by iteration 3. While some datasets exhibit minor fluctuations at intermediate iterations, the overall trend demonstrates that self-play progressively refines the model's understanding of normality. These gains validate our theoretical motivation: by iteratively contrasting real normal samples with model-generated pseudo-anomalies, DiSPaT learns a tighter description of normality that one-shot fine-tuning cannot achieve. The self-play mechanism provides the discriminative signal necessary to refine the decision boundary, enabling the model to better separate abnormal, near-normal anomalies from genuine normal instances. Additional iteration analyses for F1 and AUC-PR metrics are provided in Appendix B.6.

### 4.2. Ablation Study

**Effect of *f*-Divergence Choice.** A key feature the proposed DiSPaT framework is its inherent flexibility in choosing different *f*-divergences through Fenchel conjugate $f^*$. As discussed in Section 3, we consider functions $f$ whose conjugate $f^*$ is non-decreasing. To investigate the impact of this design choice, we evaluate four representative variants:

- **Identity:** $f^*(v) = v$
- **KL divergence:** $f^*(v) = e^v - 1$
- **Reverse KL divergence:** $f^*(v) = -\log(1 - v)$
- **Squared Hellinger distance:** $f^*(v) = \dfrac{v}{1 - v}$

Reverse KL and Squared Hellinger require $v < 1$ for well-defined conjugates, which is satisfied by applying a clamping threshold $\epsilon$ to the log-ratio before computing $f^*$. Table 4

*Table 2.* Detailed F1 scores (%) comparing DiSPaT against baselines on six-dataset benchmark. Best and second-best per dataset are highlighted in **bold** and underlined; ↑ and ↓ indicate where DiSPaT improves or degrades compared to AnoLLM with same model size. Baselines from (Tsai et al., 2025), except AnoLLM (SmolLM-1.7B) obtained via official implementation (not reported in original paper).

| Methods\Datasets | Fakejob | Lymphography | Seismic | Vehicle | 20 Newsgroups | Ecoli | Average |
|---|---|---|---|---|---|---|---|
| *Classical methods* | | | | | | | |
| Iforest | 27.4% | 23.3% | 25.1% | 11.0% | 13.7% | 75.6% | 29.4% |
| PCA | 25.6% | 56.7% | 26.6% | 12.4% | 13.3% | **77.8%** | 35.4% |
| KNN | 16.3% | 66.7% | 29.1% | 13.5% | 15.6% | **77.8%** | 36.5% |
| ECOD | 16.5% | 40.0% | 28.2% | 11.2% | 13.2% | 31.1% | 23.4% |
| *Deep Learning based methods* | | | | | | | |
| DeepSVDD | 13.6% | 56.7% | 25.8% | 11.5% | 15.2% | 53.3% | 29.4% |
| RCA | 13.7% | 66.7% | 32.0% | 13.5% | 12.9% | **77.8%** | 36.1% |
| SLAD | 17.5% | 66.7% | 28.5% | 15.5% | 15.9% | **77.8%** | 37.0% |
| GOAD | 12.9% | 66.7% | 29.5% | 11.9% | 13.6% | **77.8%** | 35.4% |
| NeuTral | 11.5% | 63.3% | 19.5% | 12.0% | 19.5% | 51.1% | 29.5% |
| ICL | 24.5% | 66.7% | 29.8% | 10.8% | 19.0% | 71.1% | 37.0% |
| DTE | 10.7% | 66.7% | 23.9% | 12.1% | 18.5% | 66.0% | 33.0% |
| REPEN | 16.4% | 66.7% | 30.6% | 12.6% | 12.4% | 75.6% | 35.7% |
| *LLM-based methods* | | | | | | | |
| AnoLLM (SmolLM-135M) | 32.5% | 76.7% | 27.9% | 16.2% | 24.1% | 33.3% | 35.1% |
| DiSPaT | 34.9%↑ | **100.0%**↑ | 33.5%↑ | **20.2%**↑ | **49.8%**↑ | 55.6%↑ | 49.0%↑ |
| AnoLLM (SmolLM-360M) | 34.3% | 80.0% | 33.6% | 17.4% | 22.0% | 33.3% | 36.8% |
| DiSPaT | **40.8%**↑ | **100.0%**↑ | **35.9%**↑ | 19.6%↓ | 40.1%↑ | **77.8%**↑ | **52.4%**↑ |
| AnoLLM (SmolLM-1.7B) | 34.4% | 83.3% | 34.1% | 16.5% | 28.1% | 44.4% | 40.1% |
| DiSPaT | 37.2%↑ | 83.3% | 32.4%↓ | 18.4%↑ | 39.2%↑ | **77.8%**↑ | 48.0%↑ |

reports results. When $f^*(v) = v$, the objective reduces to original SPIN formulation (Chen et al., 2024). While Identity improves over SFT baseline, all three $f$-divergence variants achieve higher performance, with $f^*(v) = e^v - 1$ attaining the best average scores across all metrics. These results validate our motivation for generalizing self-play to a broader family of $f$-divergences: selecting an appropriate divergence measure improves learning for anomaly detection, enabling a tighter description of normality than the Identity baseline. Derivations of $f^*$ forms and full ablation over all six datasets are provided in Appendix A.2 and B.5.

**Effect of Clamping threshold.** Figure 2 shows AUC-ROC on SmolLM-360M as the clamping threshold $\epsilon$ varies. Clamping prevents over-penalization of synthetic samples with probability ratio close to 1; $\epsilon = 0$ applies no clamping and can over-penalize "good" synthetic samples, while large $\epsilon$ reduces the effective contrast between normal and synthetic samples. Moderate positive values of $\epsilon$ yield the best performance. The results support using a moderate clamping threshold and indicate robustness to the exact value within that range. Additional analysis including F1 and AUC-PR metrics is provided in Appendix B.6.

### 4.3. Generalization Beyond Anomaly Detection

Although DiSPaT is motivated by anomaly detection, its core design is a general self-play refinement procedure.

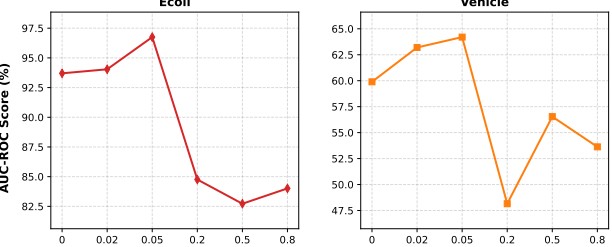

*Figure 2.* Effect of clamping threshold $\epsilon$ on AUC-ROC scores (%) on two datasets: Ecoli and Vehicle, using SmolLM-360M.

To test whether the improvement over SPIN is specific to anomaly detection or reflects a broader advantage of the objective, we additionally evaluate DiSPaT in the standard SPIN alignment setting. Following the SPIN protocol, we fine-tune Qwen1.5-1.8B on 50,000 prompts randomly sampled from UltraChat200k and evaluate on the HuggingFace Open LLM Leaderboard benchmarks.

Table 5 shows that DiSPaT consistently improves over SPIN on all six benchmarks, increasing the average score from 42.26 to 43.80. These results suggest that replacing the single fixed SPIN objective with the broader $f$-divergence family, together with the clamping mechanism, yields gains beyond the anomaly-detection setting. More broadly, they indicate that the self-play formulation of DiSPaT may be

*Table 3.* Detailed AUC-PR scores (%) comparing DiSPaT against baselines on six-dataset benchmark. Best and second-best per dataset are highlighted in **bold** and underlined; ↑ and ↓ indicate where DiSPaT improves or degrades compared to AnoLLM with same model size. Baselines from (Tsai et al., 2025), except AnoLLM (SmolLM-1.7B) obtained via official implementation (not reported in original paper).

| Methods\Datasets | Fakejob | Lymphography | Seismic | Vehicle | 20 Newsgroups | Ecoli | Average |
|---|---|---|---|---|---|---|---|
| *Classical methods* | | | | | | | |
| Iforest | 22.7% | 23.2% | 23.5% | 11.2% | 14.6% | 8.6% | 17.3% |
| PCA | 19.4% | 62.4% | 21.6% | 11.7% | 14.3% | 16.0% | 24.2% |
| KNN | 13.8% | 72.0% | 25.6% | 12.3% | 14.8% | 73.9% | 35.4% |
| ECOD | 13.0% | 36.5% | 24.4% | 11.6% | 14.1% | 18.9% | 19.8% |
| *Deep Learning based methods* | | | | | | | |
| DeepSVDD | 12.0% | 68.0% | 22.6% | 11.4% | 13.5% | 2.4% | 21.7% |
| RCA | 13.4% | 78.3% | 25.0% | 12.4% | 13.6% | 17.6% | 26.7% |
| SLAD | 15.0% | 79.5% | 24.1% | 14.0% | 15.9% | 11.0% | 26.6% |
| GOAD | 11.7% | 69.7% | 23.9% | 11.6% | 14.4% | 1.2% | 22.1% |
| NeuTral | 10.8% | 68.1% | 19.3% | 11.7% | 17.6% | 43.0% | 28.4% |
| ICL | 19.3% | 71.8% | 25.1% | 11.5% | 17.5% | 66.4% | 35.3% |
| DTE | 10.6% | 74.7% | 22.4% | 11.6% | 15.7% | **77.7%** | 35.5% |
| REPEN | 14.6% | 69.7% | 24.9% | 11.8% | 12.6% | 9.3% | 23.8% |
| *LLM-based methods* | | | | | | | |
| AnoLLM (SmolLM-135M) | 28.6% | 85.6% | 23.6% | 14.1% | 22.3% | 20.6% | 32.5% |
| DiSPaT | 30.9%↑ | **100.0%**↑ | 26.9%↑ | **17.5%**↑ | **49.4%**↑ | 53.6%↑ | 46.4%↑ |
| AnoLLM (SmolLM-360M) | 30.4% | 93.8% | 28.1% | 14.3% | 21.4% | 12.7% | 33.5% |
| DiSPaT | **36.3%**↑ | **100.0%**↑ | **29.1%**↑ | 16.9%↑ | 37.6%↑ | 65.4%↑ | **47.6%**↑ |
| AnoLLM (SmolLM-1.7B) | 31.2% | 97.6% | 27.9% | 14.3% | 26.5% | 39.4% | 39.5% |
| DiSPaT | 30.0%↓ | 97.6% | 28.3%↑ | 15.1%↑ | 36.1%↑ | 77.3%↑ | 47.4%↑ |

*Table 4.* Performance comparison for different *f*-divergence choices on four datasets using SmolLM-135M.

| $f^*$ | Metric | Fakejob | Lymph. | Vehicle | Ecoli | Avg. |
|---|---|---|---|---|---|---|
| | ROC | 84.2% | 99.3% | 59.0% | 92.3% | 83.7% |
| $v$ | F1 | 34.9% | 83.3% | 16.9% | 44.4% | 44.9% |
| | PR | 30.9% | 94.4% | 15.4% | 37.9% | 44.6% |
| | ROC | 83.9% | 100.0% | 63.8% | 94.9% | 85.7% |
| $e^v - 1$ | F1 | 35.1% | 100.0% | 20.2% | 55.6% | 52.7% |
| | PR | 30.7% | 100.0% | 17.5% | 55.6% | 50.5% |
| | ROC | 84.0% | 100.0% | 60.5% | 93.1% | 84.4% |
| $-\log(1-v)$ | F1 | 34.6% | 100.0% | 16.8% | 55.6% | 51.8% |
| | PR | 30.9% | 100.0% | 15.6% | 51.1% | 49.4% |
| | ROC | 83.8% | 97.7% | 62.8% | 94.9% | 84.8% |
| $\frac{v}{1-v}$ | F1 | 34.4% | 83.3% | 23.2% | 55.6% | 49.1% |
| | PR | 30.6% | 89.6% | 23.2% | 46.3% | 47.4% |

*Table 5.* Comparison with SPIN in the standard alignment setting. We fine-tune Qwen1.5-1.8B on 50,000 UltraChat200k prompts and evaluate on HuggingFace Open LLM Leaderboard benchmarks. Results are reported as mean ± standard deviation.

| Benchmark | SPIN | DiSPaT |
|---|---|---|
| ARC | $34.90 \pm 1.39$ | $\mathbf{36.52 \pm 1.41}$ |
| TruthfulQA | $39.59 \pm 1.45$ | $\mathbf{39.71 \pm 1.46}$ |
| Winogrande | $61.09 \pm 1.37$ | $\mathbf{62.33 \pm 1.37}$ |
| GSM8K | $29.57 \pm 1.26$ | $\mathbf{33.89 \pm 1.30}$ |
| HellaSwag | $43.66 \pm 0.49$ | $\mathbf{45.11 \pm 0.50}$ |
| MMLU | $44.75 \pm 10.05$ | $\mathbf{45.23 \pm 9.83}$ |
| Avg. | 42.26 | **43.80** |

useful in problems where informative negative or contrastive data are limited.

## 5. Conclusion

We introduce DiSPaT, a self-play fine-tuning framework for unsupervised tabular anomaly detection with LLMs. Our method uses an iterative refinement mechanism grounded in *f*-divergence minimization: at each iteration, the model generates synthetic samples that serve as pseudo-anomalies, and a discriminative objective drives the policy to better

align with the true normal data distribution. This self-play loop provides the discriminative signal absent in one-shot fine-tuning approaches, enabling the model to learn a tighter description of normality. Theoretically, we establish a tight variational approximation connecting our training objective to the *f*-divergence between real and model-induced distributions, providing a principled foundation for iterative self-improvement. Empirically, DiSPaT consistently outperforms AnoLLM, strong classical and deep-learning baselines, and additional recent tabular anomaly detection methods across multiple datasets, supporting the effectiveness of the proposed framework for tabular anomaly detection.

## Acknowledgements

Trung Le was supported by the Air Force Office of Scientific Research under award number FA2386-25-1-4023 and the ARC Discovery Project grant DP250100262. Linh Ngo Van is funded by Vietnam National Foundation for Science and Technology Development (NAFOSTED) under grant number 102.05-2025.16.

## Impact Statement

This paper presents DiSPaT, an $f$-divergence self-play fine-tuning framework for improving anomaly detection in tabular data. More accurate anomaly detection may be beneficial in applications such as fraud detection, cybersecurity, and monitoring tasks, where identifying unusual patterns reliably is important. Our work is intended to advance machine learning methodology, and we do not feel there are specific negative ethical consequences that must be highlighted here.

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

# A. Theory Development

## A.1. All Proofs

*Proof of Theorem 3.1.* We now prove (i) and (ii)

(i) By definition, we have

$$
\begin{aligned}
D_f \left( \mathbb{P}_d, \mathbb{P}_\theta \right) &= \mathbb{E}_{\mathbb{P}_\theta} \left[ f \left( \frac{p_d \left( \boldsymbol{x}, \boldsymbol{y} \right)}{p_\theta \left( \boldsymbol{x}, \boldsymbol{y} \right)} \right) \right] = \mathbb{E}_{\mathbb{P}_\theta} \left[ \max_t \left\{ \frac{p_d \left( \boldsymbol{x}, \boldsymbol{y} \right)}{p_\theta \left( \boldsymbol{x}, \boldsymbol{y} \right)} t - f^* \left( t \right) \right\} \right] \\
&\geq \max_{T \in \mathcal{T}} \mathbb{E}_{\mathbb{P}_\theta} \left[ \frac{p_d \left( \boldsymbol{x}, \boldsymbol{y} \right)}{p_\theta \left( \boldsymbol{x}, \boldsymbol{y} \right)} T \left( \boldsymbol{x}, \boldsymbol{y} \right) - f^* \left( T \left( \boldsymbol{x}, \boldsymbol{y} \right) \right) \right] \\
&= \max_{T \in \mathcal{T}} \int \left[ p_\theta \left( \boldsymbol{x}, \boldsymbol{y} \right) \frac{p_d \left( \boldsymbol{x}, \boldsymbol{y} \right)}{p_\theta \left( \boldsymbol{x}, \boldsymbol{y} \right)} T \left( \boldsymbol{x}, \boldsymbol{y} \right) - p_\theta \left( \boldsymbol{x}, \boldsymbol{y} \right) f^* \left( T \left( \boldsymbol{x}, \boldsymbol{y} \right) \right) \right] d\boldsymbol{x} d\boldsymbol{y} \\
&= \max_{T \in \mathcal{T}} \left\{ \mathbb{E}_{\mathbb{P}_d} \left[ T \left( \boldsymbol{x}, \boldsymbol{y} \right) \right] - \mathbb{E}_{\mathbb{P}_\theta} \left[ f^* \left( T \left( \boldsymbol{x}, \boldsymbol{y} \right) \right) \right] \right\}
\end{aligned}
$$

(ii) To find out the optimal $t^* = T^*(\boldsymbol{x}, \boldsymbol{y})$ for $\max_t \left\{ \frac{p_d(\boldsymbol{x},\boldsymbol{y})}{p_\theta(\boldsymbol{x},\boldsymbol{y})} t - f^* \left( t \right) \right\}$ , we consider

$$
h \left( t \right) = \frac{p_d \left( \boldsymbol{x}, \boldsymbol{y} \right)}{p_\theta \left( \boldsymbol{x}, \boldsymbol{y} \right)} t - f^* \left( t \right)
$$

Since $f$ is differentiable and strictly convex on the interior of its domain, $f^*$ is likewise differentiable with $\nabla f^* = \left( \nabla f \right)^{-1}$, and $h$ is concave; setting its derivative to zero yields

$$
h' \left( t \right) = \frac{p_d \left( \boldsymbol{x}, \boldsymbol{y} \right)}{p_\theta \left( \boldsymbol{x}, \boldsymbol{y} \right)} - \nabla f^* \left( t \right) = 0
$$

Using the property of Fenchel conjugate, we obtain

$$
t = \nabla f \left( \nabla f^* \left( t \right) \right) = \nabla f \left( \frac{p_d \left( \boldsymbol{x}, \boldsymbol{y} \right)}{p_\theta \left( \boldsymbol{x}, \boldsymbol{y} \right)} \right)
$$

which leads to

$$
T^* \left( \boldsymbol{x}, \boldsymbol{y} \right) = \nabla f \left( \frac{p_d \left( \boldsymbol{x}, \boldsymbol{y} \right)}{p_\theta \left( \boldsymbol{x}, \boldsymbol{y} \right)} \right)
$$

Consequently, we obtain

$$
\begin{aligned}
& D_f \left( \mathbb{P}_d, \mathbb{P}_\theta \right) - \mathbb{E}_{\mathbb{P}_\theta} \left[ \frac{p_d \left( \boldsymbol{x}, \boldsymbol{y} \right)}{p_\theta \left( \boldsymbol{x}, \boldsymbol{y} \right)} T \left( \boldsymbol{x}, \boldsymbol{y} \right) - f^* \left( T \left( \boldsymbol{x}, \boldsymbol{y} \right) \right) \right] \\
&= \mathbb{E}_{\mathbb{P}_\theta} \left[ \frac{p_d \left( \boldsymbol{x}, \boldsymbol{y} \right)}{p_\theta \left( \boldsymbol{x}, \boldsymbol{y} \right)} \left( T^* \left( \boldsymbol{x}, \boldsymbol{y} \right) - T \left( \boldsymbol{x}, \boldsymbol{y} \right) \right) - \left( f^* \left( T^* \left( \boldsymbol{x}, \boldsymbol{y} \right) \right) - f^* \left( T \left( \boldsymbol{x}, \boldsymbol{y} \right) \right) \right) \right] \\
&= \mathbb{E}_{\mathbb{P}_d} \left[ T^* \left( \boldsymbol{x}, \boldsymbol{y} \right) - T \left( \boldsymbol{x}, \boldsymbol{y} \right) \right] - \mathbb{E}_{\mathbb{P}_\theta} \left[ f^* \left( T^* \left( \boldsymbol{x}, \boldsymbol{y} \right) \right) - f^* \left( T \left( \boldsymbol{x}, \boldsymbol{y} \right) \right) \right] \\
&\leq \int \left| T \left( \boldsymbol{x}, \boldsymbol{y} \right) - T^* \left( \boldsymbol{x}, \boldsymbol{y} \right) \right| p_d \left( \boldsymbol{x}, \boldsymbol{y} \right) d\boldsymbol{x} d\boldsymbol{y} + \int \left| f^* \left( T \left( \boldsymbol{x}, \boldsymbol{y} \right) \right) - f^* \left( T^* \left( \boldsymbol{x}, \boldsymbol{y} \right) \right) \right| p_\theta \left( \boldsymbol{x}, \boldsymbol{y} \right) d\boldsymbol{x} d\boldsymbol{y} \\
&\leq \int \left\| T - T^* \right\|_\infty p_d \left( \boldsymbol{x}, \boldsymbol{y} \right) d\boldsymbol{x} d\boldsymbol{y} + \int \left\| f^* \circ T - f^* \circ T^* \right\|_\infty p_\theta \left( \boldsymbol{x}, \boldsymbol{y} \right) d\boldsymbol{x} d\boldsymbol{y} \\
&= \left\| T - T^* \right\|_\infty + \left\| f^* \circ T - f^* \circ T^* \right\|_\infty
\end{aligned}
$$

Therefore, we have

$$\mathbb{E}_{\mathbb{P}_\theta}\left[\frac{p_d\left(\boldsymbol{x},\boldsymbol{y}\right)}{p_\theta\left(\boldsymbol{x},\boldsymbol{y}\right)}T\left(\boldsymbol{x},\boldsymbol{y}\right)-f^*\left(T\left(\boldsymbol{x},\boldsymbol{y}\right)\right)\right]-D_f\left(\mathbb{P}_d,\mathbb{P}_\theta\right)$$

$$=\mathbb{E}_{\mathbb{P}_\theta}\left[\frac{p_d\left(\boldsymbol{x},\boldsymbol{y}\right)}{p_\theta\left(\boldsymbol{x},\boldsymbol{y}\right)}\left(T\left(\boldsymbol{x},\boldsymbol{y}\right)-T^*\left(\boldsymbol{x},\boldsymbol{y}\right)\right)-\left(f^*\left(T\left(\boldsymbol{x},\boldsymbol{y}\right)\right)-f^*\left(T^*\left(\boldsymbol{x},\boldsymbol{y}\right)\right)\right)\right]$$

$$=\mathbb{E}_{\mathbb{P}_d}\left[T\left(\boldsymbol{x},\boldsymbol{y}\right)-T^*\left(\boldsymbol{x},\boldsymbol{y}\right)\right]-\mathbb{E}_{\mathbb{P}_\theta}\left[f^*\left(T\left(\boldsymbol{x},\boldsymbol{y}\right)\right)-f^*\left(T^*\left(\boldsymbol{x},\boldsymbol{y}\right)\right)\right]$$

$$\leq\int\left|T\left(\boldsymbol{x},\boldsymbol{y}\right)-T^*\left(\boldsymbol{x},\boldsymbol{y}\right)\right|p_d\left(\boldsymbol{x},\boldsymbol{y}\right)d\boldsymbol{x}d\boldsymbol{y}+\int\left|f^*\left(T\left(\boldsymbol{x},\boldsymbol{y}\right)\right)-f^*\left(T^*\left(\boldsymbol{x},\boldsymbol{y}\right)\right)\right|p_\theta\left(\boldsymbol{x},\boldsymbol{y}\right)d\boldsymbol{x}d\boldsymbol{y}$$

$$\leq\int\left\|T-T^*\right\|_\infty p_d\left(\boldsymbol{x},\boldsymbol{y}\right)d\boldsymbol{x}d\boldsymbol{y}+\int\left\|f^*\circ T-f^*\circ T^*\right\|_\infty p_\theta\left(\boldsymbol{x},\boldsymbol{y}\right)d\boldsymbol{x}d\boldsymbol{y}$$

$$=\left\|T-T^*\right\|_\infty+\left\|f^*\circ T-f^*\circ T^*\right\|_\infty$$

This implies that

$$D_f\left(\mathbb{P}_d,\mathbb{P}_\theta\right)-\left\{\mathbb{E}_{\mathbb{P}_d}\left[T\left(\boldsymbol{x},\boldsymbol{y}\right)\right]-\mathbb{E}_{\mathbb{P}_\theta}\left[f^*\left(T\left(\boldsymbol{x},\boldsymbol{y}\right)\right)\right]\right\}\leq\left\|T-T^*\right\|_\infty+\left\|f^*\circ T-f^*\circ T^*\right\|_\infty$$

Let $\bar{T}^*=\mathrm{argmin}_{T\in\mathcal{T}}\left\|T-T^*\right\|_\infty+\left\|f^*\circ T-f^*\circ T^*\right\|_\infty$. We then have the conclusion because

$$D_f\left(\mathbb{P}_d,\mathbb{P}_\theta\right)-\max_{T\in\mathcal{T}}\left\{\mathbb{E}_{\mathbb{P}_d}\left[T\left(\boldsymbol{x},\boldsymbol{y}\right)\right]-\mathbb{E}_{\mathbb{P}_\theta}\left[f^*\left(T\left(\boldsymbol{x},\boldsymbol{y}\right)\right)\right]\right\}$$

$$\leq D_f\left(\mathbb{P}_d,\mathbb{P}_\theta\right)-\left\{\mathbb{E}_{\mathbb{P}_d}\left[\bar{T}^*\left(\boldsymbol{x},\boldsymbol{y}\right)\right]-\mathbb{E}_{\mathbb{P}_\theta}\left[f^*\left(\bar{T}^*\left(\boldsymbol{x},\boldsymbol{y}\right)\right)\right]\right\}$$

$$\leq\left\|\bar{T}^*-T^*\right\|_\infty+\left\|f^*\circ\bar{T}^*-f^*\circ T^*\right\|_\infty$$

$$=\min_{T\in\mathcal{T}}\left\{\left\|T-T^*\right\|_\infty+\left\|f^*\circ T-f^*\circ T^*\right\|_\infty\right\}$$

$\square$

*Proof of Theorem 3.2.* By definition, the Fenchel conjugate of $f$ is:

$$f^*(v)=\sup_{u\in\mathbb{R}}\{uv-f(u)\}=\sup_{u\geq 0}\{uv-g(u)\}$$

since $f(u)=+\infty$ for $u<0$ makes those terms irrelevant in the supremum. Let $h(u)=uv-g(u)$ for $u\geq 0$; since $g$ is convex, $h$ is concave with $h'(u)=v-g'(u)$, and strict convexity makes $g'$ strictly increasing from $g'_+(0)=\lim_{u\to 0^+}g'(u)=\sup\partial g(0)$ to $g'_\infty=\lim_{u\to+\infty}g'(u)$.

**Case 1:** $v\leq g'_+(0)$.

For every $u>0$, strict convexity gives $g'(u)>g'_+(0)\geq v$, so $h'(u)=v-g'(u)<0$ and $h$ is strictly decreasing. The supremum is attained at $u^*=0$:
$$f^*(v)=0\cdot v-g(0)=-g(0)$$

**Case 2:** $g'_+(0)<v<g'_\infty$.

Then $v$ lies in the range of $g'$, so $g'(u^*)=v$ has a unique solution $u^*>0$ with $h'(u^*)=0$; since $h$ is concave, $u^*$ is the maximizer:
$$f^*(v)=\sup_{u\geq 0}\{uv-g(u)\}=u^*v-g(u^*)=g^*(v)$$

**Case 3:** $v\geq g'_\infty$.

This is the boundary regime referred to in Theorem 3.2. For all $u>0$, we have $g'(u)<g'_\infty\leq v$, so $h'(u)=v-g'(u)>0$ and $h$ is strictly increasing; the supremum is therefore the limit $f^*(v)=\lim_{u\to+\infty}(uv-g(u))$. If $v>g'_\infty$, the slope is bounded below by $v-g'_\infty>0$, so this limit is $+\infty$. If $v=g'_\infty$, it equals $\lim_{u\to+\infty}(g'_\infty u-g(u))\in(-\infty,+\infty]$; hence the boundary value may be finite (e.g., $g(u)=\sqrt{u^2+1}-1$ has $g'_\infty=1$ and $f^*(1)=1$) or $+\infty$ (as for reverse KL and squared Hellinger).

**Monotonicity:** For any $v_1<v_2$ and $u\geq 0$: $uv_1\leq uv_2$, hence $uv_1-g(u)\leq uv_2-g(u)$. Taking supremum: $f^*(v_1)\leq f^*(v_2)$. $\square$

*Proof of Proposition 3.3.* We first establish the closed-form optimum of the KL-regularized problem (14), then substitute the induced critic parameterization into the critic objective (12).

Fix a context $\boldsymbol{x}$. Restricting to the support $\mathcal{Y}_{\boldsymbol{x}} = \{\boldsymbol{y}' : \pi_{\boldsymbol{\theta}_k}(\boldsymbol{y}' \mid \boldsymbol{x}) > 0\}$ (outside which any finite-objective policy must vanish, since otherwise the KL term diverges), the per-context objective in (14) reads

$$J(\pi) = \sum_{\boldsymbol{y}' \in \mathcal{Y}_{\boldsymbol{x}}} \pi(\boldsymbol{y}' \mid \boldsymbol{x}) \, T_k(\boldsymbol{x}, \boldsymbol{y}') - \beta \sum_{\boldsymbol{y}' \in \mathcal{Y}_{\boldsymbol{x}}} \pi(\boldsymbol{y}' \mid \boldsymbol{x}) \log \frac{\pi(\boldsymbol{y}' \mid \boldsymbol{x})}{\pi_{\boldsymbol{\theta}_k}(\boldsymbol{y}' \mid \boldsymbol{x})}.$$

Since the hypotheses of Theorem 3.1 ensure that $T_k$ is bounded on the support, the normalizing constant $Z(\boldsymbol{x}) = \sum_{\boldsymbol{y}' \in \mathcal{Y}_{\boldsymbol{x}}} \pi_{\boldsymbol{\theta}_k}(\boldsymbol{y}' \mid \boldsymbol{x}) \exp(\beta^{-1} T_k(\boldsymbol{x}, \boldsymbol{y}'))$ satisfies $Z(\boldsymbol{x}) \in (0, \infty)$, so the Gibbs optimum below is well defined. The reward term is linear in $\pi$ and the negative-entropy term is strictly convex, so $J$ is strictly concave; hence over the probability simplex the first-order stationarity (KKT) condition is sufficient for optimality and the constrained maximizer is unique. Introducing a multiplier $\lambda(\boldsymbol{x})$ for $\sum_{\boldsymbol{y}'} \pi(\boldsymbol{y}' \mid \boldsymbol{x}) = 1$, the Lagrangian is

$$\mathcal{L}(\pi, \lambda; \boldsymbol{x}) = J(\pi) + \lambda(\boldsymbol{x})\Big( \sum_{\boldsymbol{y}' \in \mathcal{Y}_{\boldsymbol{x}}} \pi(\boldsymbol{y}' \mid \boldsymbol{x}) - 1 \Big).$$

Differentiating with respect to a single coordinate $\pi(\boldsymbol{y}' \mid \boldsymbol{x})$ and using the product rule on the entropy term gives the stationarity condition

$$T_k(\boldsymbol{x}, \boldsymbol{y}') - \beta\Big( \log \frac{\pi(\boldsymbol{y}' \mid \boldsymbol{x})}{\pi_{\boldsymbol{\theta}_k}(\boldsymbol{y}' \mid \boldsymbol{x})} + 1 \Big) + \lambda(\boldsymbol{x}) = 0.$$

Solving for the log-ratio yields

$$\log \frac{\pi(\boldsymbol{y}' \mid \boldsymbol{x})}{\pi_{\boldsymbol{\theta}_k}(\boldsymbol{y}' \mid \boldsymbol{x})} = \beta^{-1} T_k(\boldsymbol{x}, \boldsymbol{y}') + \beta^{-1}\lambda(\boldsymbol{x}) - 1,$$

and exponentiating gives $\pi(\boldsymbol{y}' \mid \boldsymbol{x}) \propto \pi_{\boldsymbol{\theta}_k}(\boldsymbol{y}' \mid \boldsymbol{x}) \exp(\beta^{-1} T_k(\boldsymbol{x}, \boldsymbol{y}'))$, which is strictly positive on $\mathcal{Y}_{\boldsymbol{x}}$; hence the non-negativity constraints $\pi(\boldsymbol{y}' \mid \boldsymbol{x}) \geq 0$ are inactive and require no additional multiplier. Enforcing normalization determines the multiplier and yields the optimum (15),

$$\pi^*(\boldsymbol{y}' \mid \boldsymbol{x}) = \frac{1}{Z(\boldsymbol{x})} \pi_{\boldsymbol{\theta}_k}(\boldsymbol{y}' \mid \boldsymbol{x}) \exp\big(\beta^{-1} T_k(\boldsymbol{x}, \boldsymbol{y}')\big), \qquad Z(\boldsymbol{x}) = \sum_{\boldsymbol{y}' \in \mathcal{Y}_{\boldsymbol{x}}} \pi_{\boldsymbol{\theta}_k}(\boldsymbol{y}' \mid \boldsymbol{x}) \exp\big(\beta^{-1} T_k(\boldsymbol{x}, \boldsymbol{y}')\big).$$

Inverting this relation expresses the critic as an implicit function of the policy,

$$T_k(\boldsymbol{x}, \boldsymbol{y}' \mid \boldsymbol{\theta}) = \beta \log \frac{\pi_{\boldsymbol{\theta}}(\boldsymbol{y}' \mid \boldsymbol{x})}{\pi_{\boldsymbol{\theta}_k}(\boldsymbol{y}' \mid \boldsymbol{x})} + \beta \log Z(\boldsymbol{x}).$$

Substituting this parameterization for both the real score $T_k(\boldsymbol{x}, \boldsymbol{y} \mid \boldsymbol{\theta})$ and the synthetic score $T_k(\boldsymbol{x}, \boldsymbol{y}' \mid \boldsymbol{\theta})$ into the critic objective (12) gives (16). □

## A.2. Instantiations of *f*-Divergence

We present three choices of convex function $g$ that yield well-known *f*-divergences, along with their Fenchel conjugates.

### A.2.1. KL DIVERGENCE

**Definition:** $g(u) = u \log u - u + 1$ for $u > 0$, with $g(0) = 1$.

**Verification:** We verify that $D_f(P\|Q)$ recovers the KL divergence:

$$\begin{aligned} D_f(P\|Q) &= \int q(u) \cdot \left( \frac{p(u)}{q(u)} \log \frac{p(u)}{q(u)} - \frac{p(u)}{q(u)} + 1 \right) du \\ &= \int p(u) \log \frac{p(u)}{q(u)} \, du - \int p(u) \, du + \int q(u) \, du \\ &= \int p(u) \log \frac{p(u)}{q(u)} \, du - 1 + 1 = D_{KL}(P\|Q) \end{aligned}$$

**Fenchel conjugate:** We compute $g^*(v) = \sup_{u \geq 0}\{uv - g(u)\}$:

$$g^*(v) = \sup_{u \geq 0}\{uv - u \log u + u - 1\}$$

Taking the derivative and setting it to zero:

$$\frac{d}{du}(uv - u \log u + u - 1) = v - \log u = 0$$

This yields $u^* = e^v > 0$ for all $v \in \mathbb{R}$. Substituting back:

$$g^*(v) = e^v \cdot v - e^v \log(e^v) + e^v - 1 = e^v - 1$$

Since $g'(u) = \log u$, we have $g'_+(0) = \lim_{u \to 0^+} g'(u) = -\infty$ and $g'_\infty = \lim_{u \to +\infty} g'(u) = +\infty$, so the range of $g'$ is all of $\mathbb{R}$ and the interior case applies for every $v$. Thus:

$$f^*(v) = g^*(v) = e^v - 1, \quad \forall\, v \in \mathbb{R}$$

### A.2.2. REVERSE KL DIVERGENCE

**Definition:** $g(u) = -\log u + u - 1$ for $u > 0$.

**Verification:** We verify that $D_f(P\|Q)$ recovers the reverse KL divergence:

$$
\begin{aligned}
D_f(P\|Q) &= \int q(u) \cdot \left(-\log \frac{p(u)}{q(u)} + \frac{p(u)}{q(u)} - 1\right) du \\
&= -\int q(u) \log \frac{p(u)}{q(u)}\, du + \int p(u)\, du - \int q(u)\, du \\
&= \int q(u) \log \frac{q(u)}{p(u)}\, du + 1 - 1 = D_{KL}(Q\|P)
\end{aligned}
$$

**Fenchel conjugate:** We compute $g^*(v) = \sup_{u>0}\{uv - g(u)\}$:

$$g^*(v) = \sup_{u > 0}\{uv + \log u - u + 1\}$$

Taking the derivative and setting it to zero:

$$\frac{d}{du}(uv + \log u - u + 1) = v + \frac{1}{u} - 1 = 0 \quad \Rightarrow \quad u^* = \frac{1}{1 - v}$$

This requires $1 - v > 0$, i.e., $v < 1$. Substituting back:

$$g^*(v) = \frac{v}{1 - v} + \log \frac{1}{1 - v} - \frac{1}{1 - v} + 1 = -\log(1 - v)$$

Since $g'(u) = -\frac{1}{u} + 1$, we have $g'_+(0) = -\infty$ and $g'_\infty = \lim_{u \to +\infty} g'(u) = 1$, so the range of $g'$ is $(-\infty, 1)$. Thus:

$$f^*(v) = g^*(v) = -\log(1 - v)\ (v < 1), \qquad f^*(v) = +\infty\ (v \geq 1)$$

### A.2.3. SQUARED HELLINGER DISTANCE

**Definition:** $g(u) = (\sqrt{u} - 1)^2 = u - 2\sqrt{u} + 1$ for $u \geq 0$.

**Verification:** We verify that $D_f(P\|Q)$ recovers the squared Hellinger distance:

$$D_f(P\|Q) = \int q(u) \cdot \left(\sqrt{\frac{p(u)}{q(u)}} - 1\right)^2 du$$

$$= \int q(u) \cdot \left(\frac{p(u)}{q(u)} - 2\sqrt{\frac{p(u)}{q(u)}} + 1\right) du$$

$$= \int p(u)\, du - 2 \int \sqrt{p(u)q(u)}\, du + \int q(u)\, du$$

$$= 2\left(1 - \int \sqrt{p(u)q(u)}\, du\right) = H^2(P, Q)$$

**Fenchel conjugate:** We compute $g^*(v) = \sup_{u \geq 0}\{uv - g(u)\}$:

$$g^*(v) = \sup_{u \geq 0}\{uv - u + 2\sqrt{u} - 1\}$$

Let $s = \sqrt{u} \geq 0$, so $u = s^2$. The objective becomes $s^2 v - s^2 + 2s - 1$. Taking the derivative with respect to $s$:

$$\frac{d}{ds}\left(s^2 v - s^2 + 2s - 1\right) = 2s(v - 1) + 2 = 0 \quad \Rightarrow \quad s^* = \frac{1}{1-v}$$

This requires $1 - v > 0$, i.e., $v < 1$, ensuring $s^* > 0$. Thus $u^* = (s^*)^2 = \frac{1}{(1-v)^2}$. Substituting back:

$$g^*(v) = \frac{v}{(1-v)^2} - \frac{1}{(1-v)^2} + \frac{2}{1-v} - 1 = \frac{v}{1-v}$$

Since $g'(u) = 1 - \frac{1}{\sqrt{u}}$, we have $g'_+(0) = -\infty$ and $g'_\infty = \lim_{u \to +\infty} g'(u) = 1$, so the range of $g'$ is $(-\infty, 1)$. Thus:

$$f^*(v) = g^*(v) = \frac{v}{1-v} \ \ (v < 1), \qquad f^*(v) = +\infty \ \ (v \geq 1)$$

## B. Additional Experiment Result

### B.1. Implementation and Hyperparameter Details

We employ SmolLM-135M, SmolLM-360M, and SmolLM-1.7B as backbone LLMs, which are among the strongest open-weight small-scale models. All models are fine-tuned using the AdamW optimizer (Loshchilov & Hutter, 2019) with a linear warmup schedule. Training runs for 2000 steps across most datasets; Fakejob posts requires 20000 steps due to longer convergence time. The learning rate is selected from $\{1 \times 10^{-3}, 5 \times 10^{-5}\}$ per dataset. All experiments are conducted on 2×NVIDIA A100 and 2×NVIDIA L40 GPUs. For each dataset, we tune the coefficient $\beta$ and the clamping threshold $\epsilon$ via grid search over the same search space: $\{0.02, 0.05, 0.2, 0.5, 0.8\}$. We construct a validation set using 10% of the normal training samples together with an equal number of pseudo-anomalies generated by the model, and select $(\beta, \epsilon)$ by AUC-ROC on this validation set. This procedure remains fully unsupervised, as it does not use any real anomaly labels.

Following established benchmarks (Shenkar & Wolf, 2022; Xu et al., 2023), we construct training and test sets by randomly sampling 50% of normal data for training; the test set comprises the remaining normal instances and all ground-truth anomalies. For the 20 Newsgroups dataset, we follow the configuration outlined in ADBench (Han et al., 2022): we use six top-level categories (computer, recreation, science, miscellaneous, politics, and religion), conduct six experiments with each category as normal and the others as anomalies (downsampled to 5% of total instances), and report average performance across the six experiments.

We report the model performance using Area Under the Receiver Operating Characteristic Curve (AUC-ROC), F1-score and Area Under the Precision-Recall Curve (AUC-PR). Anomaly scores are computed as negative log-likelihood of serialized token sequences under the fine-tuned model, averaged over $r = 21$ random column permutations (Tsai et al., 2025).

*Table 6.* Training time and total FLOPs of AnoLLM and DiSPaT on six datasets using SmolLM-135M on a single NVIDIA A40 48GB GPU. Training time is reported in minutes.

| Dataset | AnoLLM Time | DiSPaT Time | AnoLLM FLOPs | DiSPaT FLOPs |
|---|---|---|---|---|
| Ecoli | 4.1 | 28.5 | $1.5 \times 10^{15}$ | $3.7 \times 10^{16}$ |
| Lymphography | 4.4 | 30.9 | $3.1 \times 10^{15}$ | $1.7 \times 10^{16}$ |
| 20 Newsgroups | 4.7 | 35.2 | $3.2 \times 10^{15}$ | $4.1 \times 10^{16}$ |
| Vehicle | 6.6 | 48.3 | $6.4 \times 10^{15}$ | $8.4 \times 10^{16}$ |
| Seismic | 9.7 | 67.6 | $9.8 \times 10^{15}$ | $1.2 \times 10^{17}$ |
| Fakejob | 98.8 | 671.3 | $8.4 \times 10^{16}$ | $7.7 \times 10^{17}$ |

### B.2. Training and Inference Cost

DiSPaT introduces additional training cost relative to AnoLLM because each self-play iteration evaluates log-probabilities under both the current policy and the reference policy, and trains on both real and synthetic samples. This overhead is expected and is not specific to our method; it is the standard cost of DPO/SPIN-style preference optimization. In our implementation, the per-iteration cost is approximately constant because the amount of real and synthetic data remains fixed across iterations.

Table 6 reports wall-clock training time and total FLOPs for DiSPaT and AnoLLM using SmolLM-135M on a single NVIDIA A40 48GB GPU. DiSPaT is consistently more expensive during training, with the gap becoming larger on datasets with longer serialized sequences such as Fakejob. At inference time, however, DiSPaT uses the same $r = 21$ permutation protocol as AnoLLM, so both methods incur the same inference cost when the same backbone is used. As discussed in AnoLLM, batched GPU training makes LLM-based methods practically comparable to deep tabular baselines in training time, while inference remains slower than efficient classical baselines because it requires repeated autoregressive likelihood evaluation. This trade-off can be adjusted by reducing $r$, at the cost of some accuracy. Since training is performed once offline, we view the additional training cost as acceptable given the consistent performance gains. One practical direction for reducing deployment cost is to distill a larger self-play-refined model into a smaller backbone, following recent progress in LLM distillation (Yang et al., 2024; Vuong et al., 2026).

### B.3. Additional Comparison with Recent Baselines

To address the lack of recent baselines, we additionally compare DiSPaT against five recent methods on the 21 datasets shared across all works: MCM (Yin et al., 2024), NPT-AD (Thimonier et al., 2024), DRL (Ye et al., 2025), LLM-DAS (Ye et al., 2026), and CausalTAD (Wang et al., 2026a). Results for MCM, NPT-AD, DRL, and LLM-DAS are taken directly from the reported results in LLM-DAS, while results for CausalTAD are taken directly from the reported results in CausalTAD. Table 7 reports the detailed AUC-ROC scores, along with the average score and the number of datasets on which DiSPaT achieves a higher or equal result.

DiSPaT attains the highest average AUC-ROC, outperforming MCM on 13 datasets with 2 ties, NPT-AD on 10 datasets with 2 ties, DRL on 11 datasets with 2 ties, LLM-DAS on 15 datasets, and CausalTAD on 13 datasets with 2 ties. The average AUC-ROC margin ranges from 2.0% to 12.2%, showing that DiSPaT remains competitive against recent methods from ICLR 2024, ICML 2024, ICLR 2025, and ICLR 2026.

LLM-DAS and DiSPaT use LLMs in fundamentally different ways. LLM-DAS treats the LLM as a code generator that synthesizes anomaly-detection pipelines, whereas DiSPaT uses the LLM itself as the detector and refines its generative distribution through self-play. CausalTAD is closer in starting point, as it also builds on AnoLLM, but it focuses on improving the serialization order by injecting causal knowledge and reweighting columns according to their causal importance. DiSPaT instead addresses the training paradigm itself: it follows AnoLLM's random-permutation protocol and replaces one-shot fine-tuning with iterative self-play. These directions are therefore orthogonal and potentially complementary.

### B.4. Experiment on ODDS Library

To complement the mixed-type benchmark reported in the main text, we evaluate DiSPaT on 30 additional numerical datasets from the ODDS library. Detailed per-dataset AUC-ROC, F1, and AUC-PR scores are listed in Tables 8–10. Baseline results are taken from (Tsai et al., 2025) (see table captions for exceptions and implementation notes).

*Table 7.* Detailed AUC-ROC scores (%) comparing DiSPaT against recent state-of-the-art baselines across 21 overlapping datasets. The best and second-best results in each row are highlighted in red and blue, respectively. The "Win/Tie" row indicates the number of datasets where DiSPaT outperforms or ties with the respective baseline. All baseline results are taken directly from the reported results in LLM-DAS (Ye et al., 2026) and CausalTAD (Wang et al., 2026a).

| Dataset | DiSPaT-360M | MCM | NPT-AD | DRL | LLM-DAS | CausalTAD |
|---|---|---|---|---|---|---|
| Annthyroid | **93.1** | 68.9 | 86.8 | **92.4** | 90.7 | 89.3 |
| Arr | **84.3** | 81.1 | 71.9 | 77.4 | 77.6 | **83.6** |
| Breast | **99.6** | **99.6** | 98.5 | **99.7** | 62.0 | **99.7** |
| Glass | **82.3** | 72.3 | 78.8 | 69.7 | **91.1** | 67.1 |
| Ionosphere | 92.7 | 97.3 | 97.4 | **98.4** | **99.9** | 25.2 |
| Lymphography | **100.0** | 92.6 | **99.9** | **100.0** | 90.7 | 99.3 |
| Mammography | 88.4 | 90.5 | 88.7 | **94.6** | **100.0** | 87.6 |
| Musk | **100.0** | **97.5** | **100.0** | **100.0** | 58.2 | **100.0** |
| Optdigits | 95.0 | **99.5** | 93.2 | **99.7** | 94.9 | 97.2 |
| Pendigits | 96.5 | 99.2 | **99.9** | **99.8** | 73.9 | 89.2 |
| Pima | 70.3 | **76.4** | 75.5 | **76.5** | 67.8 | 66.7 |
| Satellite | **90.2** | 79.6 | 80.6 | 83.2 | **98.4** | 84.3 |
| Satimage | **99.9** | **99.9** | **100.0** | 99.7 | 99.8 | **100.0** |
| Shuttle | **100.0** | 99.8 | 99.3 | **99.9** | 40.2 | **100.0** |
| Speech | 48.2 | 44.1 | 59.3 | 58.2 | **98.5** | **62.5** |
| Thyroid | **99.3** | 98.0 | 97.9 | **99.1** | 19.2 | 97.6 |
| Vertebral | **80.4** | 37.7 | 53.5 | 62.2 | 55.9 | **63.1** |
| Vowels | 95.5 | 65.3 | **99.4** | 85.1 | **96.6** | 73.1 |
| WBC | 95.5 | **98.1** | 96.2 | **99.6** | 93.3 | 96.9 |
| Wine | **96.2** | 95.4 | **96.2** | **100.0** | 45.6 | **100.0** |
| Yeast | **80.3** | 42.6 | 46.9 | 51.2 | **76.7** | 54.3 |
| **Average** | **89.9** | 82.6 | 86.6 | **87.9** | 77.7 | 82.7 |
| **Win/Tie** | - | 13/2 | 10/2 | 11/2 | 15/0 | 13/2 |

Across these benchmarks, DiSPaT frequently attains top or near-top performance on individual datasets and yields strong average scores across metrics and model scales. The results indicate that the proposed self-play refinement generalizes beyond mixed-type tables and remains competitive on predominantly numerical tasks, supporting the robustness claims made in the Experiments 4. While performance varies across datasets—reflecting inherent differences in data characteristics and anomaly patterns—the consistent improvements over AnoLLM across all three model scales validate the effectiveness of iterative self-play refinement for numerical tabular data.

### B.5. Extended $f$-Divergence Ablation

Table 11 extends the $f^*$-ablation of Section 4.2 to all six datasets using SmolLM-135M. The full benchmark corroborates the main-text findings: KL divergence ($f^*(v) = e^v - 1$) leads on average across AUC-ROC, F1, and AUC-PR, while Identity ($f^*(v) = v$) lags; Reverse KL and Squared Hellinger sit between them. This ordering is preserved on Seismic and 20 Newsgroups—the advantage of $f$-divergence variants over the Identity (SPIN) objective thus holds across the full benchmark. While isolated per-dataset differences exist—e.g., Squared Hellinger achieves the best F1 on Vehicle insurance—these do not alter the overall ranking, supporting the design choice of KL divergence as the default $f$-divergence in DiSPaT.

### B.6. Iteration Curves and Clamping Threshold $\epsilon$

**Iteration Curves.** Figure 3 shows F1 and AUC-PR scores of DiSPaT (SmolLM-135M) over self-play iterations on four datasets. We observe that all datasets improve over the SFT baseline within a few iterations, though the shape of the curve differs—some rise steadily, others dip then recover. The optimal iteration is dataset-dependent, and several runs peak before the final iteration, which supports using a small, fixed number of iterations as a default and suggests that validation-based iteration selection may be useful in practice. The convergence patterns reflect the iterative refinement process: initial improvements come from learning to distinguish synthetic pseudo-anomalies, while later iterations may show diminishing returns as the model's understanding of normality stabilizes.

*Table 8.* Detailed AUC-ROC scores comparing DiSPaT against baselines over 30 datasets in ODDS. The best and second-best results in each row are indicated in red and blue, respectively. All baseline results are reported directly from (Tsai et al., 2025).

| Datasets | Classical Methods | | | | Deep-learning based Methods | | | | | | | | LLM-based Methods | | | | | |
| | Iforest | PCA | KNN | ECOD | DeepSVDD | RCA | SLAD | GOAD | NeuTral | ICL | DTE | REPEN | 135M | | 360M | | 1.7B | |
| | | | | | | | | | | | | | AnoLLM | DiSPaT | AnoLLM | DiSPaT | AnoLLM | DiSPaT |
| Annthyroid | 92.2 | 83.9 | 81.1 | 79.0 | 74.2 | 71.8 | 76.1 | 57.2 | 81.3 | 84.2 | **97.7** | 73.6 | 92.7 | 92.9 | 93.1 | 93.1 | 93.0 | **94.0** |
| Arrhythmia | **82.7** | 79.6 | 78.6 | 81.1 | 76.5 | 78.6 | 78.4 | 68.1 | 76.0 | 78.5 | 77.1 | 68.4 | 82.5 | 81.4 | 82.2 | **84.3** | 82.4 | 82.4 |
| BreastW | **99.4** | 98.8 | 99.2 | 99.2 | 97.4 | 98.7 | 98.6 | **99.4** | 98.3 | 99.2 | 98.2 | 95.5 | 99.2 | **99.6** | 99.3 | **99.6** | 99.1 | **99.4** |
| Cardio | 94.8 | **96.6** | 92.1 | 93.5 | 84.2 | 94.8 | 84.0 | 52.4 | 85.9 | 89.4 | 92.0 | 82.9 | 94.0 | **94.9** | 87.3 | 91.7 | 86.7 | 87.7 |
| Ecoli | 85.6 | 85.6 | 87.7 | 77.6 | 88.7 | 88.3 | 88.2 | 88.1 | 86.0 | 88.7 | 82.1 | 87.0 | 77.7 | 94.9 | 80.4 | **96.8** | 79.1 | **95.1** |
| ForestCover | 87.0 | 94.5 | **98.5** | 92.1 | 53.3 | 94.4 | 85.7 | 27.8 | 89.8 | 97.7 | **97.8** | 90.2 | 88.1 | 92.2 | 83.5 | 91.3 | 88.7 | 88.9 |
| Glass | 80.2 | 71.3 | 84.9 | 69.3 | 82.4 | 71.9 | 79.0 | 57.4 | **93.3** | **88.7** | 79.9 | 75.5 | 81.9 | 82.6 | 79.7 | 82.3 | 81.8 | 82.5 |
| Heart | 81.9 | **84.2** | 81.4 | 65.9 | 77.3 | 78.5 | 82.2 | **83.8** | 81.1 | 78.7 | 83.1 | 44.9 | 82.0 | 79.7 | 79.9 | 80.7 | 80.3 | 83.3 |
| Http | 99.2 | **99.9** | **100.0** | 97.9 | 99.0 | 99.5 | **99.9** | 99.6 | 97.3 | **100.0** | 99.5 | 99.4 | **100.0** | **100.0** | **100.0** | **100.0** | **100.0** | **100.0** |
| Ionosphere | 89.1 | 89.4 | 96.0 | 73.4 | 96.3 | 91.6 | 96.0 | 95.0 | 95.6 | **97.0** | **96.4** | 54.5 | 90.9 | **96.4** | 92.4 | 92.7 | 91.8 | 93.0 |
| LETTER | 63.1 | 52.9 | 86.5 | 56.7 | 77.6 | 71.6 | 90.8 | 81.1 | 92.9 | **95.9** | 87.2 | 59.7 | **96.7** | 83.7 | 86.7 | 80.2 | 77.2 | 84.2 |
| Lymphography | 67.3 | 82.6 | 86.0 | 83.0 | 89.9 | 91.9 | 96.4 | 81.7 | 84.7 | 82.7 | 90.9 | 80.8 | 96.8 | **100.0** | 99.3 | **100.0** | **99.5** | **100.0** |
| Mammography | 88.1 | 90.0 | 87.2 | **90.6** | 85.7 | 87.3 | 74.0 | 75.6 | 69.0 | 78.2 | 86.4 | 86.3 | **91.5** | 87.2 | 87.6 | 88.4 | 87.4 | 88.1 |
| Mulcross | **99.9** | **100.0** | **100.0** | 96.0 | **100.0** | **100.0** | 96.9 | **100.0** | 96.8 | **100.0** | **100.0** | 97.3 | **100.0** | **100.0** | **100.0** | **100.0** | **100.0** | **100.0** |
| Musk | 97.1 | **100.0** | **100.0** | 95.6 | **100.0** | **100.0** | **100.0** | **100.0** | **100.0** | **100.0** | **100.0** | 72.2 | **100.0** | **100.0** | **100.0** | **100.0** | **100.0** | **100.0** |
| Optdigits | 82.4 | 57.4 | 94.4 | 60.6 | 75.3 | 80.0 | 73.5 | 84.7 | **97.9** | 95.5 | 88.8 | 60.7 | **98.3** | 94.5 | 93.9 | 95.0 | 88.8 | 95.9 |
| Pendigits | 97.1 | 94.2 | **99.9** | 92.8 | 83.8 | 96.7 | 93.2 | 22.0 | 96.3 | 97.1 | **98.2** | 92.2 | 97.1 | 97.7 | 96.4 | 96.5 | 93.0 | 96.0 |
| Pima | 72.0 | 71.1 | **74.1** | 58.7 | 59.3 | 70.4 | 58.4 | 66.5 | **76.3** | 69.7 | 66.2 | 68.8 | 66.3 | 70.9 | 65.4 | 70.3 | 65.3 | 72.1 |
| Satellite | 80.7 | 66.2 | 87.4 | 58.2 | 80.0 | 73.4 | 86.4 | 79.5 | 85.9 | 85.4 | 78.9 | 74.0 | **90.2** | 88.7 | 87.7 | **90.2** | 85.8 | **89.8** |
| Satimage-2 | 99.3 | 97.9 | **99.9** | 96.5 | 96.1 | 99.8 | 99.7 | 99.4 | 86.1 | 99.7 | 98.8 | 99.8 | **100.0** | **99.9** | **99.9** | **99.9** | **99.9** | **100.0** |
| Seismic | 69.2 | 69.2 | 73.8 | 69.2 | 71.3 | 72.7 | 71.4 | 71.7 | 68.1 | 71.9 | 71.4 | 72.4 | 71.2 | 73.2 | **74.6** | **75.7** | 74.0 | **74.6** |
| Shuttle | 99.6 | 99.4 | **99.9** | 99.3 | 99.7 | 99.6 | 99.0 | 99.0 | 99.7 | **99.9** | 99.8 | 99.2 | **100.0** | **99.9** | **100.0** | **99.9** | **99.9** | **100.0** |
| Smtp | 90.5 | 80.9 | **93.6** | 88.0 | 78.0 | 84.5 | 92.6 | 91.1 | 89.0 | 88.5 | **95.3** | 89.4 | 92.7 | 92.1 | 92.9 | 91.9 | 92.4 | 93.0 |
| Speech | 47.8 | 47.1 | 48.6 | 47.1 | 56.3 | 47.2 | 51.1 | 55.0 | **60.9** | **58.2** | 51.1 | 53.1 | 47.0 | 48.1 | 47.0 | 48.2 | 47.0 | 47.4 |
| Thyroid | 98.9 | 98.4 | 97.6 | 97.8 | 86.9 | 96.9 | 94.8 | 68.9 | 88.6 | 98.7 | 99.0 | 90.4 | 97.5 | 97.7 | 98.3 | **99.3** | **99.1** | **99.3** |
| Vertebral | 44.6 | 49.4 | 40.6 | 47.4 | 47.8 | 47.8 | 48.3 | 51.6 | 54.5 | 54.3 | 57.0 | 24.7 | 56.5 | **63.9** | 40.8 | **80.4** | 39.2 | 52.2 |
| Vowels | 77.9 | 64.4 | 97.5 | 59.7 | 89.5 | 89.1 | 96.9 | 92.5 | **98.8** | 97.9 | **98.3** | 75.3 | 98.2 | 96.0 | 93.8 | 95.5 | 93.3 | 94.9 |
| WBC | 94.7 | 94.9 | 94.7 | 90.7 | 90.4 | 94.6 | 92.8 | 66.3 | 76.5 | 90.8 | 91.3 | 77.0 | **96.4** | 94.8 | 95.2 | 95.5 | **95.7** | 95.0 |
| Wine | 93.0 | 92.7 | 95.2 | 72.9 | 85.4 | 89.9 | 96.4 | 96.1 | 96.0 | 94.4 | **97.4** | 83.5 | 90.9 | **97.2** | 85.1 | 96.2 | 87.6 | 96.0 |
| Yeast | **86.3** | **85.0** | 80.2 | 78.7 | 70.6 | 81.6 | 41.6 | 23.5 | 62.9 | 71.3 | 78.1 | 69.9 | 74.4 | 77.8 | 73.0 | 80.3 | 75.4 | 75.7 |
| Average | 84.7 | 82.6 | 87.9 | 79.0 | 81.8 | 84.8 | 84.1 | 74.5 | 85.5 | 87.7 | 87.9 | 76.6 | 88.4 | **89.4** | 86.5 | **90.5** | 86.1 | **89.4** |

**Clamping Threshold.** Figure 4 reports F1 and AUC-PR on Ecoli and Vehicle (SmolLM-360M) as the clamping threshold $\epsilon$ varies. Clamping restricts the log-ratio so that synthetic samples with probability ratio close to 1 are not over-penalized as pseudo-anomalies; $\epsilon = 0$ applies no clamping and can therefore over-penalize "good" synthetic samples, while overly large $\epsilon$ shrinks the effective contrast between normal and synthetic samples in the objective. The figure shows that moderate positive values of $\epsilon$ yield the best performance and that large $\epsilon$ degrades it. The results support the use of a moderate clamping threshold in the main setup and indicate that the method is robust to the exact value within that range.

### B.7. Dataset Information

Table 12 summarizes all datasets used in this work. Datasets are drawn from the AnoLLM benchmark (Tsai et al., 2025) and the ODDS library (Rayana, 2016), which may include textual, numerical, or categorical features. Columns report the number of samples, counts of textual features, numerical features, and categorical features, and the number and percentage of anomalies.

### B.8. Examples of Model-Generated Samples

We present examples from three datasets: Fakejob (mixed: textual, numerical, categorical), Ecoli (numerical), and Vehicle (mixed: numerical, categorical). For each sample, we first draw a permutation $\pi \in S_d$ and serialize the row accordingly. We then randomly partition the serialized features into *context* and *target* sets. The context features are preserved as the input $x$, while the target features form the continuation $y$ to be generated sequentially. Concretely, the model predicts target features using prompts of the form "`feature is value, ...`", conditioning on the preserved context and on previously generated target features. Because $\pi$ is resampled during training, both the feature ordering and the induced context-continuation split vary across permutations. As a result, the same feature may appear in either the context or the continuation under different permutations, encouraging the model to learn dependencies across many possible orderings.

This variation can be seen directly in Figure 5. In the Fakejob example, iteration 0 uses `benefits`, `telecommuting`, and `has questions` as targets; iteration 1 uses `company profile`, `industry`, and `salary range`; and iteration 2 uses `description`, `function`, and `telecommuting`. The preserved context changes accordingly. The same principle

*Table 9.* Detailed F1 scores comparing DiSPaT against baselines over 30 datasets in ODDS. The best and second-best results in each row are indicated in red and blue, respectively. Baseline results are reported from (Tsai et al., 2025), except for AnoLLM (SmolLM-1.7B) which is obtained using the official implementation as they were not available in the original report.

| Datasets | Classical Methods | | | | Deep-learning based Methods | | | | | | | | LLM-based Methods | | | | | |
|---|---|---|---|---|---|---|---|---|---|---|---|---|---|---|---|---|---|---|
| | | | | | | | | | | | | | 135M | | 360M | | 1.7B | |
| | Iforest | PCA | KNN | ECOD | DeepSVDD | RCA | SLAD | GOAD | NeuTral | ICL | DTE | REPEN | AnoLLM | DiSPaT | AnoLLM | DiSPaT | AnoLLM | DiSPaT |
| Annthyroid | 57.4 | 48.7 | 44.0 | 38.8 | 43.6 | 36.7 | 41.8 | 25.7 | 46.8 | 50.1 | 78.9 | 33.8 | 58.4 | 59.2 | 59.7 | 62.0 | 62.6 | 64.6 |
| Arrhythmia | 61.2 | 54.2 | 55.4 | 59.1 | 53.3 | 54.2 | 53.6 | 50.3 | 51.5 | 53.3 | 52.1 | 45.2 | 61.2 | 60.4 | 60.0 | 62.1 | 62.1 | 60.7 |
| BreastW | 96.9 | 95.9 | 96.3 | 95.4 | 93.6 | 95.9 | 95.1 | 96.6 | 96.7 | 95.9 | 96.3 | 93.5 | 95.8 | 97.1 | 96.6 | 97.1 | 96.7 | 96.7 |
| Cardio | 71.5 | 80.8 | 67.6 | 66.6 | 56.4 | 72.6 | 60.2 | 29.4 | 56.8 | 68.9 | 64.4 | 56.1 | 73.4 | 72.2 | 66.5 | 65.9 | 63.4 | 64.2 |
| Ecoli | 75.6 | 77.8 | 77.8 | 31.1 | 53.3 | 77.8 | 77.8 | 77.8 | 51.1 | 71.1 | 66.0 | 75.6 | 33.3 | 55.6 | 33.3 | 77.8 | 35.6 | 77.8 |
| ForestCover | 10.9 | 15.8 | 74.5 | 23.8 | 3.5 | 18.9 | 13.6 | 0.1 | 42.6 | 76.9 | 77.8 | 6.4 | 25.6 | 27.1 | 21.3 | 24.0 | 18.6 | 20.2 |
| Glass | 15.6 | 13.3 | 17.8 | 15.6 | 24.4 | 15.6 | 17.8 | 13.3 | 42.2 | 28.9 | 13.3 | 6.7 | 17.8 | 21.1 | 17.8 | 22.2 | 15.6 | 22.2 |
| Heart | 91.7 | 92.2 | 90.6 | 89.3 | 90.6 | 90.8 | 91.3 | 91.0 | 90.4 | 91.0 | 91.4 | 87.6 | 91.2 | 91.4 | 90.7 | 91.5 | 90.9 | 91.0 |
| Http (KDDCUP99) | 10.7 | 92.6 | 99.4 | 2.2 | 45.9 | 38.2 | 92.9 | 43.8 | 19.3 | 99.3 | 34.9 | 20.4 | 98.9 | 97.0 | 95.8 | 96.2 | 93.4 | 98.2 |
| Ionosphere | 79.7 | 78.9 | 89.4 | 66.0 | 89.5 | 83.2 | 89.4 | 85.4 | 88.2 | 90.6 | 90.0 | 59.5 | 82.1 | 89.7 | 83.8 | 84.4 | 84.3 | 85.7 |
| Letter Recognition | 17.6 | 13.6 | 43.4 | 14.6 | 40.4 | 29.0 | 54.8 | 40.4 | 63.6 | 72.2 | 58.8 | 16.4 | 73.4 | 66.2 | 48.6 | 47.4 | 32.0 | 40.1 |
| Lymphography | 23.3 | 56.7 | 66.7 | 40.0 | 56.7 | 66.7 | 66.7 | 66.7 | 63.3 | 66.7 | 66.7 | 66.7 | 76.7 | 100.0 | 80.0 | 100.0 | 83.3 | 83.3 |
| Mammography | 41.3 | 47.4 | 40.9 | 53.5 | 44.3 | 35.8 | 13.8 | 28.7 | 13.5 | 29.8 | 36.4 | 29.4 | 55.1 | 44.2 | 42.8 | 45.4 | 41.5 | 42.5 |
| Mulcross | 99.5 | 100.0 | 100.0 | 74.7 | 100.0 | 99.9 | 76.0 | 100.0 | 85.2 | 99.6 | 100.0 | 81.6 | 100.0 | 100.0 | 100.0 | 100.0 | 100.0 | 100.0 |
| Musk | 61.6 | 100.0 | 100.0 | 54.6 | 100.0 | 100.0 | 100.0 | 100.0 | 100.0 | 100.0 | 100.0 | 15.0 | 100.0 | 100.0 | 100.0 | 100.0 | 100.0 | 100.0 |
| Optdigits | 15.9 | 0.1 | 28.4 | 2.7 | 29.7 | 2.1 | 0.0 | 16.5 | 63.9 | 47.1 | 16.4 | 2.0 | 72.0 | 50.7 | 44.3 | 46.7 | 34.0 | 63.3 |
| Pendigits | 55.1 | 44.2 | 91.0 | 42.7 | 44.5 | 53.0 | 35.6 | 0.0 | 61.0 | 60.6 | 37.3 | | 55.9 | 57.1 | 50.5 | 52.6 | 36.7 | 56.4 |
| Pima | 67.2 | 68.8 | 69.2 | 57.8 | 57.0 | 67.2 | 58.5 | 62.8 | 69.5 | 67.0 | 62.4 | 66.8 | 62.6 | 64.6 | 62.0 | 66.0 | 61.7 | 68.3 |
| Satellite | 69.6 | 61.4 | 76.2 | 53.8 | 71.0 | 68.5 | 76.0 | 69.3 | 75.1 | 75.7 | 72.3 | 69.1 | 79.8 | 79.8 | 77.4 | 80.3 | 75.3 | 79.8 |
| Satimage-2 | 87.3 | 84.8 | 93.5 | 71.8 | 88.4 | 94.9 | 82.5 | 95.5 | 5.1 | 91.8 | 50.1 | 16.4 | 95.2 | 94.4 | 94.4 | 94.4 | 92.6 | 95.8 |
| Seismic | 25.1 | 26.6 | 29.1 | 28.2 | 25.8 | 32.0 | 28.5 | 29.5 | 19.5 | 29.9 | 23.9 | 30.6 | 27.9 | 33.5 | 31.6 | 35.9 | 34.1 | 32.4 |
| Shuttle | 96.4 | 95.8 | 97.7 | 91.7 | 98.3 | 96.9 | 97.5 | 96.5 | 98.2 | 98.3 | 97.4 | 93.6 | 98.3 | 98.1 | 98.4 | 97.8 | 98.4 | 98.4 |
| Smtp (KDDCUP99) | 0.0 | 66.7 | 66.7 | 66.7 | 9.3 | 65.3 | 66.7 | 66.7 | 60.7 | 49.3 | 66.7 | 60.7 | 66.7 | 65.4 | 66.7 | 66.7 | 66.7 | 65.0 |
| Speech | 3.0 | 4.9 | 5.6 | 4.9 | 3.6 | 4.9 | 3.9 | 3.9 | 4.9 | 7.5 | 5.2 | 1.3 | 6.6 | 6.6 | 6.2 | 6.6 | 6.6 | 6.6 |
| Thyroid | 78.9 | 72.3 | 64.3 | 62.6 | 55.7 | 60.0 | 63.9 | 41.7 | 35.0 | 77.8 | 80.4 | 34.6 | 68.2 | 78.5 | 72.5 | 89.2 | 79.6 | 81.7 |
| Vertebral | 18.7 | 20.7 | 14.0 | 21.3 | 28.0 | 16.0 | 16.0 | 30.0 | 32.7 | 19.3 | 30.0 | 1.3 | 28.7 | 30.0 | 18.0 | 46.7 | 16.7 | 33.3 |
| Vowels | 24.8 | 20.0 | 68.4 | 22.0 | 54.8 | 42.4 | 71.6 | 46.0 | 76.8 | 74.0 | 78.0 | 24.4 | 76.0 | 64.8 | 55.6 | 60.0 | 60.0 | 66.0 |
| WBC | 81.9 | 79.0 | 71.4 | 56.2 | 70.5 | 72.4 | 66.7 | 38.1 | 18.1 | 73.3 | 61.9 | 21.9 | 79.0 | 78.4 | 76.2 | 85.7 | 71.4 | 76.2 |
| Wine | 70.0 | 62.0 | 70.0 | 38.0 | 42.0 | 60.0 | 72.0 | 76.0 | 70.0 | 62.0 | 72.0 | 44.0 | 50.0 | 80.0 | 52.0 | 70.0 | 62.0 | 70.0 |
| Yeast | 48.6 | 42.1 | 33.7 | 33.5 | 31.0 | 35.2 | 8.4 | 1.0 | 18.1 | 33.1 | 32.8 | 15.8 | 32.0 | 35.8 | 31.0 | 39.0 | 29.9 | 42.1 |
| Average | 51.9 | 57.2 | 64.8 | 46.0 | 53.5 | 56.2 | 56.6 | 50.8 | 53.5 | 65.1 | 61.2 | 43.0 | 64.7 | 66.6 | 61.1 | 67.8 | 60.2 | 66.1 |

applies to the Ecoli and Vehicle examples below, where different sampled permutations expose different subsets of features as targets at each iteration.

Figure 5 shows iterative refinement on mixed features (Fakejob). At iteration 0, the model generates oversimplified values (e.g., benefits = "$11") or incorrect ones (e.g., has questions = -1.0). By iteration 1, values become more structured but lack detail. By iteration 2, some features are correct while others remain generic.

Figure 6 shows results on numerical features (Ecoli). At iteration 0, the model produces malformed values with formatting errors (e.g., "0 ." with trailing punctuation). By iteration 1, values are properly formatted but magnitudes are incorrect. By iteration 2, formatting is correct but subtle discrepancies remain.

Figure 7 shows results on mixed numerical and categorical features (Vehicle). At iteration 0, categorical values are incorrect (e.g., "3Sum" instead of "Sedan - Collision") and numerical values are malformed. By iteration 1, categorical values improve but errors persist. By iteration 2, most categorical values are correct while numerical discrepancies remain.

The sequential approach preserves column names, ensures coherence through conditional generation, and enables iterative refinement via self-play. Examples show progressive improvement from clearly anomalous values at iteration 0 to more realistic but still distinguishable values at later iterations.

## C. Related Works

Tabular anomaly detection has been extensively studied, with methods ranging from classical statistical approaches to recent deep learning models. Classical methods include isolation forest (Liu et al., 2008), empirical-cumulative-distribution-based outlier detector (ECOD) (Li et al., 2022), and $k$-nearest neighbors (KNN) (Ramaswamy et al., 2000). Earlier kernel-based novelty-detection research developed explicit geometric data descriptions around the normal region, including hypersphere models with separately adjustable inner and outer margins for imbalanced settings (Le et al., 2010), multiple spherical boundaries for multi-modal normal data (Le et al., 2011), semi-supervised SVDD with fuzzy memberships and entropy for partially labeled data (Le et al., 2013), and cutting-plane accelerations for scalable one-class SVM training (Le et al., 2015).

*Table 10.* Detailed AUC-PR scores comparing DiSPaT against baselines over 30 datasets in ODDS. The best and second-best results in each row are indicated in red and blue, respectively. Baseline results are reported from (Tsai et al., 2025), except for AnoLLM (SmolLM-1.7B) which is obtained using the official implementation as they were not available in the original report.

| Datasets | Classical Methods | | | | Deep-learning based Methods | | | | | | | | LLM-based Methods | | | | | |
|---|---|---|---|---|---|---|---|---|---|---|---|---|---|---|---|---|---|---|
| | | | | | | | | | | | | | 135M | | 360M | | 1.7B | |
| | Iforest | PCA | KNN | ECOD | DeepSVDD | RCA | SLAD | GOAD | NeuTral | ICL | DTE | REPEN | AnoLLM | DiSPaT | AnoLLM | DiSPaT | AnoLLM | DiSPaT |
| Annthyroid | 64.6 | 55.0 | 46.3 | 40.6 | 44.1 | 38.3 | 46.1 | 28.6 | 43.5 | 55.5 | 83.5 | 34.3 | 63.1 | 64.4 | 64.8 | 69.0 | 69.4 | 70.8 |
| Arrhythmia | 66.2 | 61.7 | 55.6 | 62.2 | 56.3 | 56.2 | 55.6 | 51.8 | 50.7 | 55.6 | 56.6 | 41.9 | 63.6 | 65.1 | 64.2 | 70.3 | 60.2 | 63.7 |
| BreastW | 99.4 | 98.5 | 99.2 | 99.2 | 96.6 | 98.6 | 98.3 | 99.4 | 97.0 | 99.1 | 96.7 | 92.5 | 99.1 | 99.6 | 99.2 | 99.6 | 99.2 | 99.5 |
| Cardio | 78.6 | 84.4 | 73.7 | 71.2 | 60.6 | 74.5 | 66.7 | 32.5 | 61.0 | 75.0 | 67.8 | 56.7 | 81.1 | 79.1 | 72.6 | 72.2 | 69.2 | 69.6 |
| Ecoli | 8.6 | 16.0 | 73.9 | 18.9 | 2.4 | 17.6 | 11.0 | 1.2 | 43.0 | 66.4 | 77.7 | 9.3 | 20.6 | 53.6 | 12.7 | 65.4 | 39.3 | 77.3 |
| ForestCover | 64.9 | 71.0 | 78.6 | 30.6 | 62.1 | 73.9 | 72.8 | 79.6 | 47.5 | 80.7 | 58.3 | 68.2 | 41.9 | 43.9 | 41.7 | 43.6 | 43.5 | 44.5 |
| Glass | 19.8 | 16.7 | 24.2 | 24.2 | 26.3 | 18.7 | 20.8 | 15.0 | 48.4 | 37.4 | 22.6 | 16.5 | 24.7 | 21.1 | 23.4 | 24.7 | 25.2 | 34.7 |
| Heart | 97.2 | 97.6 | 97.2 | 94.0 | 96.4 | 96.6 | 97.3 | 97.7 | 97.2 | 96.3 | 97.5 | 88.4 | 97.2 | 96.2 | 96.9 | 96.4 | 97.0 | 97.6 |
| Http (KDDCUP99) | 49.6 | 91.1 | 99.5 | 25.4 | 58.8 | 40.3 | 91.0 | 61.8 | 36.4 | 99.5 | 58.3 | 53.6 | 97.0 | 96.0 | 95.6 | 96.0 | 92.7 | 97.5 |
| Ionosphere | 89.8 | 91.2 | 96.7 | 76.9 | 96.7 | 93.2 | 96.9 | 95.8 | 95.9 | 97.7 | 97.2 | 53.4 | 93.3 | 97.3 | 93.2 | 94.5 | 94.2 | 94.9 |
| Letter Recognition | 16.8 | 14.3 | 42.6 | 14.1 | 41.2 | 25.9 | 57.8 | 39.0 | 70.3 | 77.3 | 55.0 | 15.1 | 79.7 | 50.2 | 19.1 | 46.9 | 29.2 | 45.7 |
| Lymphography | 23.2 | 62.4 | 72.0 | 36.5 | 68.0 | 78.3 | 79.5 | 69.7 | 68.1 | 71.8 | 74.7 | 69.7 | 85.6 | 100.0 | 93.8 | 100.0 | 97.6 | 97.6 |
| Mammography | 39.2 | 44.3 | 39.9 | 54.8 | 44.7 | 31.2 | 12.6 | 23.2 | 9.4 | 28.7 | 37.8 | 26.8 | 59.2 | 45.1 | 36.4 | 46.9 | 39.1 | 43.6 |
| Mulcross | 98.9 | 100.0 | 100.0 | 72.2 | 100.0 | 100.0 | 78.8 | 100.0 | 81.6 | 99.8 | 100.0 | 100.0 | 100.0 | 100.0 | 100.0 | 100.0 | 100.0 | 100.0 |
| Musk | 66.6 | 100.0 | 100.0 | 62.7 | 100.0 | 100.0 | 100.0 | 100.0 | 100.0 | 100.0 | 100.0 | 17.5 | 100.0 | 100.0 | 100.0 | 100.0 | 100.0 | 100.0 |
| Optdigits | 16.6 | 5.9 | 31.4 | 6.5 | 23.2 | 12.2 | 10.9 | 17.8 | 64.5 | 41.4 | 22.2 | 7.6 | 75.0 | 58.1 | 39.8 | 43.5 | 28.8 | 66.4 |
| Pendigits | 54.4 | 37.6 | 95.8 | 39.5 | 41.6 | 51.6 | 29.2 | 2.6 | 40.8 | 65.6 | 50.9 | 31.9 | 62.3 | 67.7 | 55.4 | 56.9 | 42.5 | 63.4 |
| Pima | 71.4 | 69.6 | 71.6 | 62.2 | 60.6 | 69.8 | 60.3 | 66.0 | 74.6 | 69.5 | 63.9 | 67.3 | 67.7 | 70.1 | 67.4 | 68.2 | 66.3 | 70.4 |
| Satellite | 84.5 | 76.9 | 88.9 | 65.8 | 84.2 | 80.6 | 86.6 | 80.8 | 86.0 | 88.7 | 84.3 | 80.6 | 91.0 | 90.3 | 89.1 | 91.5 | 87.1 | 91.1 |
| Satimage-2 | 93.0 | 90.1 | 98.0 | 74.5 | 95.0 | 97.7 | 90.3 | 98.0 | 8.2 | 96.7 | 52.6 | 95.2 | 98.8 | 98.1 | 97.4 | 98.1 | 96.5 | 98.8 |
| Seismic | 23.5 | 21.6 | 25.6 | 24.4 | 22.6 | 25.0 | 24.1 | 23.9 | 19.3 | 25.0 | 22.4 | 24.9 | 23.6 | 26.9 | 28.1 | 29.1 | 27.9 | 28.3 |
| Shuttle | 98.4 | 96.2 | 97.2 | 94.6 | 98.7 | 96.0 | 96.8 | 94.9 | 99.4 | 99.5 | 94.6 | 92.8 | 99.7 | 98.5 | 99.6 | 98.7 | 99.5 | 99.7 |
| Smtp (KDDCUP99) | 1.0 | 45.4 | 45.9 | 60.8 | 5.8 | 44.1 | 46.9 | 44.1 | 58.2 | 37.7 | 46.7 | 40.3 | 65.8 | 62.2 | 64.5 | 57.6 | 59.0 | 63.6 |
| Speech | 3.5 | 3.7 | 3.8 | 4.0 | 4.2 | 3.7 | 3.6 | 4.0 | 4.0 | 5.7 | 4.0 | 3.5 | 3.6 | 4.0 | 3.7 | 6.6 | 3.7 | 4.0 |
| Thyroid | 78.3 | 79.1 | 69.6 | 63.5 | 56.0 | 64.9 | 68.6 | 40.1 | 33.0 | 82.2 | 86.0 | 38.5 | 69.6 | 82.6 | 74.0 | 92.8 | 84.7 | 87.9 |
| Vertebral | 21.0 | 23.2 | 19.2 | 22.8 | 25.2 | 21.4 | 21.0 | 28.1 | 30.3 | 26.4 | 31.0 | 15.1 | 28.9 | 30.0 | 18.1 | 47.7 | 22.2 | 31.7 |
| Vowels | 22.9 | 16.2 | 76.2 | 15.3 | 60.3 | 45.5 | 76.5 | 54.4 | 86.1 | 80.4 | 83.1 | 20.3 | 83.9 | 64.7 | 59.9 | 60.0 | 61.4 | 68.0 |
| WBC | 84.2 | 87.6 | 81.4 | 58.6 | 74.7 | 80.8 | 71.1 | 40.8 | 22.6 | 71.4 | 64.0 | 23.5 | 87.3 | 83.5 | 75.3 | 89.0 | 75.8 | 81.4 |
| Wine | 67.2 | 65.9 | 71.1 | 32.1 | 51.2 | 51.7 | 78.2 | 78.9 | 77.9 | 73.4 | 87.3 | 48.4 | 52.2 | 87.3 | 52.9 | 78.1 | 68.1 | 80.3 |
| Yeast | 44.0 | 34.6 | 29.4 | 32.3 | 29.9 | 30.3 | 10.6 | 7.6 | 16.8 | 26.2 | 28.2 | 18.7 | 30.1 | 29.9 | 30.2 | 34.3 | 26.9 | 35.0 |
| Average | 54.9 | 58.6 | 66.8 | 48.0 | 56.1 | 57.3 | 58.7 | 52.6 | 55.8 | 67.7 | 63.5 | 44.4 | 68.2 | 68.8 | 62.3 | 69.3 | 63.5 | 70.2 |

Deep-learning-based approaches include margin-based methods such as DeepSVDD (Ruff et al., 2018) and self-supervised learning methods such as REPEN (Pang et al., 2018) and SLAD (Xu et al., 2023). More recent tabular anomaly detectors have substantially strengthened this line of work. MCM (Yin et al., 2024) extends masked modeling to tabular anomaly detection and learns multiple masking patterns to capture diverse feature correlations. NPT-AD (Thimonier et al., 2024) leverages both feature-feature and sample-sample dependencies through a non-parametric transformer-based reconstruction framework. DRL (Ye et al., 2025) improves discrimination by remapping samples into a decomposed latent space designed to separate shared normal patterns from anomalous variation. These methods are strong tabular baselines, but they are primarily developed for conventional tabular representations and do not directly exploit the generative and semantic capabilities of LLMs on heterogeneous rows containing text, categorical values, and numerical attributes.

Large language models (LLMs) have recently become an attractive foundation for mixed-type tabular anomaly detection because they can operate on a serialized representation of each row and naturally exploit the semantics of categorical values and raw text (Dinh et al., 2022; Hegselmann et al., 2023). Previous work demonstrates that LLMs can effectively handle tabular prediction tasks through text serialization (Dinh et al., 2022), while few-shot classification on tabular data benefits from serialization strategies that preserve semantic relationships (Hegselmann et al., 2023). AnoLLM (Tsai et al., 2025) applies LLMs to tabular anomaly detection by serializing rows into text templates, discretizing numerical values, and fine-tuning LLMs on normal data via next-token prediction. LLM-DAS (Ye et al., 2026) uses LLMs in a different way: rather than treating the LLM as the detector itself, it uses the LLM as an algorithmist to synthesize detector-specific anomaly-generation programs that strengthen downstream anomaly detectors. CausalTAD (Wang et al., 2026a) stays closer to the AnoLLM setting, but focuses on improving the serialization order by injecting causal knowledge and reweighting columns according to their causal importance. In contrast, DiSPaT addresses the training paradigm itself. Rather than relying on one-shot likelihood maximization on normal data, it introduces iterative self-play to refine the model's description of normality through progressively harder model-generated not-normal samples. This distinction is important because our goal is not to redesign the detector architecture or the serialization order, but to strengthen the learning signal available to LLM-based tabular anomaly detection.

Our theoretical framework builds upon foundational works in divergence minimization and self-play optimization. The

*Table 11.* Performance comparison of SmolLM-135M with different $f$-divergence choices across datasets. Metrics reported are AUC-ROC, F1-score, and AUC-PR.

| $f^*$ | Metric | Fakejob | Lymphography | Vehicle insurance | Ecoli | Seismic | 20 Newsgroups | Average |
|---|---|---|---|---|---|---|---|---|
| $v$ | AUC-ROC | 84.2% | 99.3% | 59.0% | 92.3% | 73.2% | 91.3% | 83,2% |
| | F1 | 34.9% | 83.3% | 16.9% | 44.4% | 32.4% | 46.4% | 43.1% |
| | AUC-PR | 30.9% | 94.4% | 15.4% | 37.9% | 26.6% | 46.7% | 42.0% |
| $e^v - 1$ | AUC-ROC | 83.9% | 100.0% | 63.8% | 94.9% | 73.2% | 91.9% | 84.6% |
| | F1 | 35.1% | 100.0% | 20.2% | 55.6% | 33.5% | 49.4% | 49.0% |
| | AUC-PR | 30.7% | 100.0% | 17.5% | 53.6% | 26.9% | 49.8% | 46.4% |
| $\dfrac{v}{1-v}$ | AUC-ROC | 84.0% | 100.0% | 60.5% | 93.1% | 73.0% | 91.9% | 83.8% |
| | F1 | 34.6% | 100.0% | 16.8% | 55.6% | 32.9% | 48.6% | 48.1% |
| | AUC-PR | 30.9% | 100.0% | 15.6% | 51.1% | 27.0% | 48.6% | 45.5% |
| $-\log(1-v)$ | AUC-ROC | 83.8% | 97.7% | 62.8% | 94.9% | 73.1% | 91.8% | 84.0% |
| | F1 | 34.4% | 83.3% | 23.2% | 55.6% | 33.5% | 48.6% | 46.4% |
| | AUC-PR | 30.6% | 89.6% | 23.2% | 46.3% | 26.4% | 48.1% | 44.0% |

framework of $f$-divergences provides a general framework for measuring distributional differences (Csiszár & Shields, 2004; Liese & Vajda, 2006). GAN objectives can be generalized to arbitrary $f$-divergences through variational lower bounds (Nowozin et al., 2016). We extend this framework to the conditional setting where distributions depend on context, providing explicit approximation bounds. Self-play mechanisms have been successfully applied to language model alignment. SPIN (Chen et al., 2024) introduces a self-play fine-tuning approach where models learn by distinguishing their own generations from ground-truth data, providing a learning signal without external annotations. Direct Preference Optimization (DPO) (Rafailov et al., 2023) simplifies preference learning by directly optimizing a binary classification objective. Related ideas on strengthening alignment or mitigating representation bias under indirect or limited supervision also appear in other settings, including neural topic modeling and few-shot continual relation extraction (Vuong et al., 2025; Pham et al., 2025). The idea of using synthetic samples as negative examples has been explored in semi-supervised anomaly detection. Synthetic anomalies can be generated via data augmentation (Ruff et al., 2020), while deviation networks learn from both normal and synthetic anomalous samples (Pang et al., 2019). Synthetic anomalies can also be generated through adversarial perturbations (Ding et al., 2022). We adapt these mechanisms to unsupervised tabular anomaly detection by minimizing $f$-divergence between normal data and model-generated distributions, where self-play generates pseudo-anomalies that serve as negative examples, enabling unsupervised learning without requiring anomaly labels or external generators.

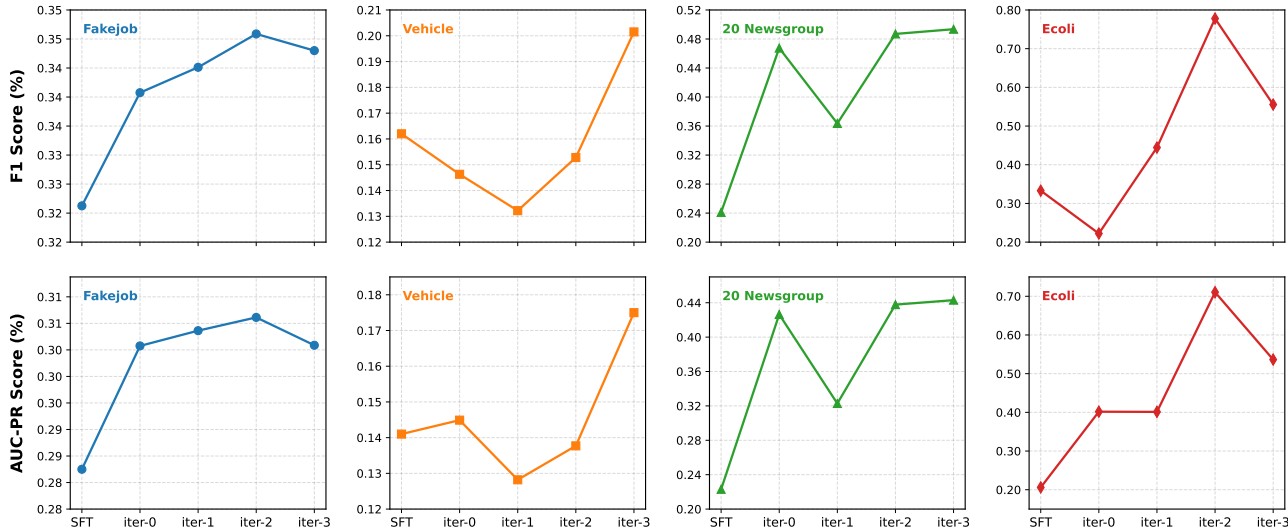

*Figure 3.* F1 scores (%) and AUC-PR scores (%) of DiSPaT across self-play iterations on four datasets using SmolLM-135M. SFT indicates initial model, and iter-$k$ refers to model after $k$ iterations of training.

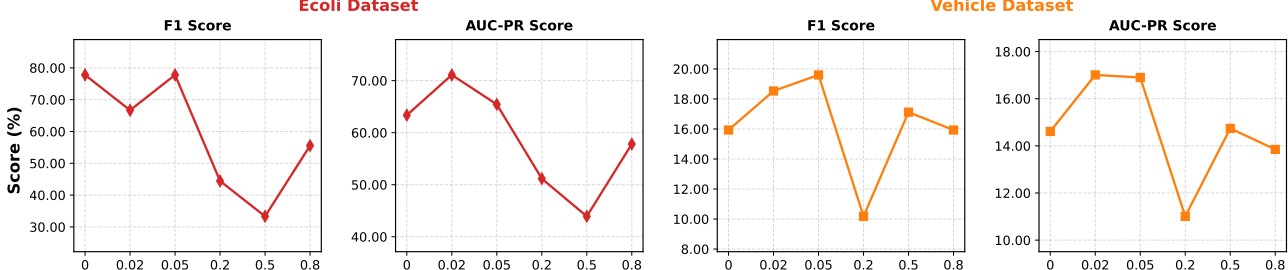

*Figure 4.* Effect of clamping threshold $\epsilon$ on F1 scores (%) and AUC-PR scores (%) on two datasets using SmolLM-360M.

*Table 12.* Statistics for all datasets used in this work. Datasets are drawn from the AnoLLM benchmark (Tsai et al., 2025) and the ODDS library (Rayana, 2016).

| Dataset | Samples | Textual | Numerical | Categorical | Anomalies (%) |
|---|---|---|---|---|---|
| 20 Newsgroup | 11905 | 1 | 0 | 0 | 591 (5.0%) |
| Annthyroid | 7200 | 0 | 6 | 0 | 534 (7.42%) |
| Arrhythmia | 452 | 0 | 274 | 0 | 66 (15%) |
| BreastW | 683 | 0 | 9 | 0 | 239 (35%) |
| Cardio | 1831 | 0 | 21 | 0 | 176 (9.6%) |
| Ecoli | 336 | 0 | 7 | 0 | 9 (2.6%) |
| Fakejob | 17880 | 5 | 3 | 8 | 866 (4.8%) |
| ForestCover | 286048 | 0 | 10 | 0 | 2747 (0.9%) |
| Glass | 214 | 0 | 9 | 0 | 9 (4.2%) |
| Heart | 224 | 0 | 44 | 0 | 10 (4.4%) |
| Http (KDDCUP99) | 567479 | 0 | 3 | 0 | 2211 (0.4%) |
| Ionosphere | 351 | 0 | 33 | 0 | 126 (36%) |
| Letter Recognition | 1600 | 0 | 32 | 0 | 100 (6.25%) |
| Lymphography | 148 | 0 | 3 | 15 | 6 (4.1%) |
| Mammography | 11183 | 0 | 6 | 0 | 260 (2.32%) |
| Mulcross | 262144 | 0 | 4 | 0 | 26214 (10%) |
| Musk | 3062 | 0 | 166 | 0 | 97 (3.2%) |
| Optdigits | 5216 | 0 | 64 | 0 | 150 (3%) |
| Pendigits | 6870 | 0 | 16 | 0 | 156 (2.27%) |
| Pima | 768 | 0 | 8 | 0 | 268 (35%) |
| Satellite | 6435 | 0 | 36 | 0 | 2036 (32%) |
| Satimage-2 | 5803 | 0 | 36 | 0 | 71 (1.2%) |
| Seismic | 2584 | 0 | 14 | 4 | 170 (6.6%) |
| Shuttle | 49097 | 0 | 9 | 0 | 3511 (7%) |
| Smtp (KDDCUP99) | 95156 | 0 | 3 | 0 | 30 (0.03%) |
| Speech | 3686 | 0 | 400 | 0 | 61 (1.65%) |
| Thyroid | 3772 | 0 | 6 | 0 | 93 (2.5%) |
| Vehicle | 15420 | 0 | 8 | 24 | 923 (6.0%) |
| Vertebral | 240 | 0 | 6 | 0 | 30 (12.5%) |
| Vowels | 1456 | 0 | 12 | 0 | 50 (3.4%) |
| WBC | 278 | 0 | 30 | 0 | 21 (5.6%) |
| Wine | 129 | 0 | 13 | 0 | 10 (7.7%) |
| Yeast | 1364 | 0 | 8 | 0 | 64 (4.7%) |

---

**Prompt:** Given context features (job posting attributes), generate target features sequentially.

### Iteration 0:

**Prompt:** `benefits` is [generate], `telecommuting` is [generate], `has questions` is [generate].

`benefits:` Work With Us and Get... Responsibility: What you do here will have a direct impact on our growth. Flexible working hours: We focus on results, not punching a clock. Paid time off: You get 15 vacation days and 12 company holidays. Location: Centrally located in Bangalore's tallest building... Perks: Business Cards & T-Shirts, Sodexo Meal Pass, Gym Membership, Movie ticket + Pizza twice a month... [Additional detailed perks]
`telecommuting:` -0.2
`has questions:` 1.0

`benefits:` $11
`telecommuting:` -0.2
`has questions:` -1.0

*(a)* Chosen: ground truth (normal sample)    *(b)* Rejected: model generation @Iter 0

### Iteration 1:

**Prompt:** `company profile` is [generate], `industry` is [generate], `salary range` is [generate].

`company profile:` Who are we? Creators Lab is a young creative Branding, Web Design and Digital Marketing Startup based in Bangalore. We are specially focused on helping startups with their branding and design needs. We are also building few revolutionary products. One of them is EnterBox, its Roku for India. We focus on: Brand Strategy, Web Design & Development, UI/UX Design, Mobile Applications, SEO and Social Media Marketing. [Additional details about team and work culture]
`industry:` Information Technology and Services
`salary range:` 120000-300000

`company profile:` Planning Centre
`industry:` Computer Games
`salary range:` 140000-240000

*(c)* Chosen: ground truth (normal sample)    *(d)* Rejected: model generation @Iter 1

### Iteration 2:

**Prompt:** `description` is [generate], `function` is [generate], `telecommuting` is [generate].

`description:` You're the one we're looking for if... You have incredible design talent. Can design an awesome logo, business card, posters and UI for an app, etc. You believe that good design is about much more than just 1px bevels and subtle gradients. It's all about problem solving. General Purpose: We're looking for a user interface / experience designer to take care of design needs of our client projects and our products. Main Job Tasks: Brainstorm and mock-up ideas, Creating UI concepts with paper sketches, wireframes, Translate business needs into user-focused experiences... [Additional detailed responsibilities]
`function:` Design
`telecommuting:` -0.2

`description:` the leading electronic and communications business for the Small and Medium Enterprise business segment around the world
`function:` Design
`telecommuting:` -0.2

*(e)* Chosen: ground truth (normal sample)    *(f)* Rejected: model generation @Iter 2

*Figure 5.* Iterative refinement of synthetic samples through self-play (Fakejob dataset). At iteration 0, the model generates severely oversimplified values (e.g., `benefits` = "$11" instead of a detailed description) and incorrect values (e.g., `has questions` = -1.0 instead of 1.0), clearly marking them as pseudo-anomalies. By iteration 1, the model generates structured values but lacks detail and context coherence (e.g., `company profile` = "Planning Centre" vs. a detailed company description; `industry` = "Computer Games" which mismatches the Design job context). By iteration 2, some features are correctly generated (e.g., `function`, `telecommuting`), but textual fields remain generic rather than job-specific. This progression demonstrates how self-play enables the model to learn increasingly subtle distinctions between synthetic and real normal samples.

**Prompt:** Given context features (protein sequence scores), generate target features sequentially.

**Iteration 0:**

**Prompt:** `Presence of charge on N-terminus of predicted lipoproteins is` [generate]`, von Heijne's method for signal sequence recognition is` [generate]`, McGeoch's method for signal sequence recognition is` [generate].

```
Presence of charge on N-terminus of
predicted lipoproteins: -0.1
von Heijne's method for signal sequence
recognition: -0.5
McGeoch's method for signal sequence
recognition: 2.0
```

```
Presence of charge on N-terminus of
predicted lipoproteins: 0 .
von Heijne's method for signal sequence
recognition: 0.1 . Both partial least squares...
McGeoch's method for signal sequence
recognition: 0.7 .
```

*(a)* Chosen: ground truth (normal sample)      *(b)* Rejected: model generation @Iter 0

**Iteration 1:**

**Prompt:** `score of ALOM program after excluding putative cleavable signal regions from the sequence is` [generate]`, score of the ALOM membrane spanning region prediction program is` [generate]`, von Heijne's Signal Peptidase II consensus sequence score is` [generate].

```
score of ALOM program after excluding
putative cleavable signal regions from
the sequence: 1.2
score of the ALOM membrane spanning region
prediction program: 1.1
von Heijne's Signal Peptidase II consensus
sequence score: -0.2
```

```
score of ALOM program after excluding
putative cleavable signal regions from
the sequence: 1.7
score of the ALOM membrane spanning region
prediction program: 1.1
von Heijne's Signal Peptidase II consensus
sequence score: 0.5
```

*(c)* Chosen: ground truth (normal sample)      *(d)* Rejected: model generation @Iter 1

**Iteration 2:**

**Prompt:** `von Heijne's Signal Peptidase II consensus sequence score is` [generate]`, score of discriminant analysis of the amino acid content of outer membrane and periplasmic proteins is` [generate]`, score of ALOM program after excluding putative cleavable signal regions from the sequence is` [generate].

```
von Heijne's Signal Peptidase II consensus
sequence score: -0.2
score of discriminant analysis of the
amino acid content of outer membrane and
periplasmic proteins: 0.2
score of ALOM program after excluding
putative cleavable signal regions from
the sequence: 1.2
```

```
von Heijne's Signal Peptidase II consensus
sequence score: -0.2
score of discriminant analysis of the
amino acid content of outer membrane and
periplasmic proteins: 1.8
score of ALOM program after excluding
putative cleavable signal regions from
the sequence: 1.7
```

*(e)* Chosen: ground truth (normal sample)      *(f)* Rejected: model generation @Iter 2

*Figure 6.* Iterative refinement on numerical features (Ecoli dataset). At iteration 0, the model generates malformed numerical values with formatting errors (e.g., "0 ." with trailing punctuation, or "0.1 . Both partial least squares..." with appended text), clearly marking them as pseudo-anomalies. By iteration 1, the model generates properly formatted numbers but with incorrect magnitudes (e.g., 1.7 instead of 1.2, 0.5 instead of -0.2). By iteration 2, while formatting is correct, subtle numerical discrepancies persist (e.g., 1.8 instead of 0.2, 1.7 instead of 1.2), demonstrating progressive refinement in learning numerical distributions.

**Prompt:** Given context features (insurance claim attributes), generate target features sequentially.

**Iteration 0:**

**Prompt:** `age of vehicle is` [generate], `policy type is` [generate], `day of week is` [generate], `week of month claimed is` [generate].

`age of vehicle:` 5 years
`policy type:` Sedan - Collision
`day of week:` Wednesday
`week of month claimed:` 0.2

`age of vehicle:` 1 to 2 years
`policy type:` 3Sum
`day of week:` 0.4
`week of month claimed:` 0.6

*(a)* Chosen: ground truth (normal sample)    *(b)* Rejected: model generation @Iter 0

**Iteration 1:**

**Prompt:** `agent type is` [generate], `policy type is` [generate], `age of vehicle is` [generate], `number of cars is` [generate].

`agent type:` External
`policy type:` Sedan - Collision
`age of vehicle:` 5 years
`number of cars:` 1 vehicle

`agent type:` Male
`policy type:` No
`age of vehicle:` 50
`number of cars:` 1 vehicle

*(c)* Chosen: ground truth (normal sample)    *(d)* Rejected: model generation @Iter 1

**Iteration 2:**

**Prompt:** `age of vehicle is` [generate], `week of month claimed is` [generate], `week of month is` [generate], `age of policy holder is` [generate].

`age of vehicle:` 5 years
`week of month claimed:` 0.2
`week of month:` -0.6
`age of policy holder:` 31 to 35

`age of vehicle:` 7 years
`week of month claimed:` 0.2
`week of month:` 0.2
`age of policy holder:` 31 to 35

*(e)* Chosen: ground truth (normal sample)    *(f)* Rejected: model generation @Iter 2

*Figure 7.* Iterative refinement on mixed categorical and numerical features (Vehicle dataset). At iteration 0, the model generates incorrect categorical values (e.g., "3Sum" instead of "Sedan - Collision" for policy type, "1 to 2 years" instead of "5 years" for age of vehicle) and malformed numerical values (e.g., "0.4" instead of "Wednesday" for day of week, "0.6" instead of 0.2 for week of month claimed), clearly marking them as pseudo-anomalies. By iteration 1, categorical values improve but still contain errors (e.g., "No" instead of "Sedan - Collision" for policy type, "50" instead of "5 years" for age of vehicle, "Male" instead of "External" for agent type). By iteration 2, most categorical values are correct, but subtle numerical discrepancies persist (e.g., "7 years" instead of "5 years" for age of vehicle, "0.2" instead of -0.6 for week of month), demonstrating progressive refinement in learning both categorical and numerical distributions.

