# OpenReview forum: "$f$-Divergence Self-Play for Tabular Anomaly Detection via Large Language Models"
_ICML.cc/2026/Conference — ICML 2026 regular_

### Official Review · Reviewer_p89C · 2026-03-03

**Soundness:** 2
**Presentation:** 2
**Significance:** 2
**Originality:** 2
**Overall Recommendation:** 3
**Confidence:** 4

**Summary:**

This paper proposes DiSPaT, a self-play fine-tuning framework for unsupervised tabular anomaly detection using LLMs. The method serializes mixed-type rows as text and iteratively refines an LLM by contrasting real normal samples with the model’s own generations, framed as minimizing an f-divergence between the empirical normal-data distribution and the model-induced distribution via a variational approximation.

**Compliance With Llm Reviewing Policy:**

Affirmed.

**Final Justification:**

After considering the rebuttal and the perspectives of the other reviewers, I maintain my current score.

**Key Questions For Authors:**

In addition to the comments in the weaknesses section, I also have the following concerns:
1. Both the baseline NTP training and the proposed self-play method operate in an unsupervised setting with respect to anomaly labels (i.e., no normal/abnormal supervision is used during training). Therefore, the improvement cannot be attributed to a stronger supervision signal, but rather to a different optimization dynamic.
2. The x–y decomposition for “context–continuation” pairs is under-specified for serialized rows; it is not clear how contexts are formed for generation and whether they vary across permutations.
3. Hyperparameter selection is said to be “based on performance on the training set” in an uncontaminated unsupervised setting. It is unclear what metric was used without labels. If any labeled anomalies or test labels were consulted, this would be leakage; if not, a description of the unsupervised selection criterion is needed.

**Limitations:**

No. While the paper focuses primarily on methodological development and empirical performance, it provides limited discussion of potential limitations and broader societal impacts. In particular, the authors could elaborate on risks related to false positives/negatives in anomaly detection, potential misuse in high-stakes domains, and biases introduced by training solely on normal data.

**Strengths And Weaknesses:**

Strengths

1. Extends self-play fine-tuning (SPIN) to unsupervised tabular AD with a principled f-divergence variational formulation.
2. Mathematical framing connects to standard variational f-divergences (f-GAN style), aiding interpretability of the objective family.
3. DiSPaT improves over AnoLLM and a suite of classical and deep-learning baselines, with ablations examining divergence choices, self-play iterations, and a clamping mechanism.

Weaknesses

1. The proposed self-play framework appears conceptually similar to existing adversarial or self-play fine-tuning paradigms. As such, the core training mechanism does not seem fundamentally novel, but rather an adaptation of established self-play / f-divergence minimization frameworks to tabular anomaly detection. The paper would benefit from a clearer articulation of what is genuinely new beyond the application.
2. Compared to standard next-token prediction (NTP) fine-tuning, the proposed method introduces iterative generation, divergence-based optimization, and additional hyperparameters. However, the paper does not provide sufficient analysis on training cost, convergence behavior, or sensitivity to these additional components.
3. Inconsistency between the stated alternating critic–policy game and the final training algorithm: Algorithm 1 implements an implicit-critic, SPIN-like single update using log-ratio terms, but the paper repeatedly describes a learned discriminator. The role and training of an explicit critic remain unclear.
4. The derivation from the min–max in Eq. (7) to the implementable loss in Eq. (17) relies on a particular parameterization of T via log-ratios; however, the precise assumptions ensuring equivalence (and stability when f*(·) is non-linear and both terms depend on θ) are not fully justified.
5. Although the method is positioned as an LLM-based anomaly detection approach, the experimental comparison includes only one LLM-based baseline (AnoLLM), while the majority of baselines are classical and deep-learning baselines methods. Including additional LLM-based variants or stronger LLM-centric baselines would provide a more convincing evaluation.
6. Limited analysis of decoding/generation details for y′ and their impact on pseudo-anomaly quality and training stability.
7. Typos/formatting issues occasionally impede flow. For example, the acronym Self-Play fIne-tuNing (SPIN) should be properly defined upon its first occurrence. In addition, the sentence spanning lines 129–130 appears to require a comma rather than a period.

---

> ### Author Rebuttal · Authors · 2026-03-30
>
> Dear Reviewer p89C, **with extended comparison against recent SOTA methods (ICML '24, ICLR '24–'26), DiSPaT achieves state-of-the-art with 2.0–12.2% AUC-ROC margin over all baselines. Framework also outperforms on LLM alignment (https://anonymous.4open.science/r/rebuttal_icml26/Alignment.png), provide versatility beyond AD. Our framework is validated by rigorous theory, consistent empirical gains. Supplementary comparison: https://anonymous.4open.science/r/rebuttal_icml26/Result.png**. We detail our responses below.
>
> **W1**: Contribution extends well beyond applying SPIN to new domain:
> - SPIN optimizes one fixed IPM-like divergence. We **generalize to family of $f$-divergences**, letting practitioners select divergence suited to data characteristics—critical for AD where normal manifolds vary across domains.
> - **Theoretical development is non-trivial**: Theorem 3.1 gives tight variational bounds; Theorem 3.3 derives closed-form objective. $f$-DPO similarly generalizes DPO to $f$-divergences, recognized at ICLR 2024 for analogous theoretical contributions.
> - Clamping (Eq.18) **handles a challenge absent in SPIN**: penalizing synthetic samples already close to normal data is counterproductive.
> - **Empirically**, Tab.4 confirms $f$-divergence variants outperform SPIN baseline. We evaluated SPIN alignment (Qwen1.5-1.8B, 50K UltraChat200k). **DiSPaT outperforms SPIN on all benchmarks**, confirm practical gains beyond AD:
> |Method|ARC|TruthfulQA|Winogrande|GSM8K|HellaSwag|MMLU|Avg|
> |-|-|-|-|-|-|-|-|
> |SPIN|34.90|39.59|61.09|29.57|43.66|44.75|42.26|
> |DiSPaT|**36.52**|**39.71**|**62.33**|**33.89**|**45.11**|**45.23**|**43.80**|
>
> **W2**: Sensitivity and convergence are already analyzed in paper.
> - Sensitivity: Sec. 4.2, Appendix B.3-B.4 present ablations: $f$-divergence (Tab. 4,8), $\epsilon$ (Fig.2,4), iteration depth (Fig.1,3). Moderate $\epsilon$ is best, $f$-divergence variants beat SPIN, most datasets improve in 2-3 iterations.
> - Convergence: Fig.1,3 (Sec.4.1, Appendix B.4) show gains over SFT within a few iterations, mostly monotonic or near-monotonic. KL regularization keeps updates bounded.
> - Training cost: detailed times/FLOPs are in our response to **Reviewer xhes (W1+Q2)**. Higher cost is standard for DPO/SPIN dual-model forward passes, not specific to DiSPaT. Inference cost matches AnoLLM.
>
> **W3**: Follow DPO/SPIN, critic is updated implicitly. From Eq.(14), optimal policy (Eq.15) is $$\pi^\ast(y'|x)=\frac{1}{Z(x)}\pi_{\theta_k}(y'|x)\exp\left(\beta^{-1}T_k(x,y')\right)$$
> Critic is thus implicit function of $\theta$: $$T_k(x,y'|\theta)=\beta\log\frac{\pi_\theta(y'|x)}{\pi_{\theta_k}(y'|x)}+\beta\log Z(x)$$ Substitute into Eq.(12) gives Eq.(16), then Eq.(17). Each iteration has one $\theta$ update jointly realizing critic and policy updates (Alg. 1).
>
> **W4**: Derivation from Eq.(7) to Eq.(17) uses explicit assumptions detailed in paper:
> - $f$ in Theorem 3.2, so $f^*$ is non-decreasing (Appendix A.2).
> - Expressivity of $\mathcal{T}$: LLMs are powerful function approximators, making gap $d(T^*,\mathcal{T})$ negligible in practice.
> - Eq.(14) admits closed-form Eq.(15), well-known in KL-regularized policy optimization.
> - From Eq.(16) to Eq.(17): $\beta\log Z(x)$ constant w.r.t. $\theta$; $f^*$ non-decreasing (Theorem 3.2), so equivalence is exact.
> - Stability: logistic loss (Sec. 2.3) is smooth, with bounded gradients (discuss after Eq.(12)). KL regularization constrains updates close to $\pi_{\theta_k}$. Fig.1,3 confirm stable convergence.
>
> **W5**: Per Reviewer 1wTh, we compare against recent SOTA baselines (ICML '24, ICLR '24-'26) **using results directly reported in LLM-DAS (covering MCM, DRL, NPT-AD) and CausalTAD**. DiSPaT's 2.0%-12.2% average AUC-ROC margin and consistent win/tie rates strongly validate our approach:
> |Metric|DiSPaT|MCM|NPT-AD|DRL|LLM-DAS|CausalTAD|
> |-|-|-|-|-|-|-|
> |Avg AUC-ROC|**89.9%**|82.6%|86.6%|87.9%|77.7%|82.7%|
> |W/T|-|13/2|10/2|11/2|15/0|13/2|
>
> **W6**: Following AnoLLM serialization, model generates $y'\sim\pi_{\theta_k}(\cdot|x)$ conditioned on context $x$ (Alg. 1, Sec. 2.1; Appendix B.6). Fig.5-7 show pseudo-samples move from abnormal toward near-normal across iterations, and Fig.1,3 show performance improve then plateau. This indicates generated samples are informative for boundary refinement.
>
> **Q1**: Standard NTP optimizes only $\log p_\theta(y|x)$ on normal data; once dominant patterns are learned, gradients weaken. DiSPaT adds explicit contrastive signal: maximize likelihood on $y$ while penalizing generated $y'$. This keeps persistent signal to tighten normality boundary. Fig.1 confirms after SFT convergence, self-play still improves detection.
>
> **Q2**: Context-continuation follows AnoLLM serialization (Sec. 2.1; Appendix B.6). Features are split into context and target per permutation; each feature appears in both roles, avoiding overfitting to fixed ordering and improving dependency learning.
>
> **Q3**: Please see our response to **Reviewer xhes (W2+Q3)** for full protocol.

---

> > ### Author Rebuttal · Reviewer_p89C · 2026-04-04
> >
> > I appreciate the effort to clarify the contributions and provide further empirical evidence. The rebuttal provides additional positioning, which helps clarify differences. However, the core concern remains. The method still appears conceptually close to existing frameworks. The rebuttal improves justification but does not fully establish fundamentally new learning principles beyond these extensions. Additionally, the analysis of training cost, stability across hyperparameters, and practical overhead is still limited and not sufficiently quantified in a clear and comparable manner.

---

> > > ### Author Response · Authors · 2026-04-05
> > >
> > > We thank reviewer's feedback and clarify the remaining points.
> > >
> > > **On novelty.** We maintain that DiSPaT is a principled generalization with distinct motivation, theory, and design—not an application of existing frameworks to a new domain.
> > >
> > > Standard SFT maximizes $\log p_\theta(y|x)$ on labeled data. However, once dominant corpus regularities are learned, further training inherently plateaus, yielding diminishing returns or degradation. Self-play was proposed to address this: SPIN optimizes $\max_{f\in\mathcal{F}_t}\mathbb{E}[f(x,y)-f(x,y')]$ over function class $\mathcal{F}_t$—a single, fixed **Integral Probability Metrics (IPM)** objective, where model distinguishes its own generations $y'$ from ground-truth $y$, pushing quality beyond what SFT alone can achieve. Our approach departs from a different starting point. We minimize $D_f(P\_d,P\_\theta)\approx\max\_{T\in\mathcal{T}}\\{\\mathbb{E}\_{P\_d}[T(x,y)]-\mathbb{E}\_{P\_\theta}[f^\ast(T(x,y'))]\\}$ over function family $\mathcal{T}$, where $f^\*$ is Fenchel conjugate of convex generator $f$, giving rise to **family of divergences** through different choices of $f$. SPIN is recovered when $f^*(v)=v$ (identity), and Tab. 4 confirms other choices consistently outperform it. We also evaluated DiSPaT in standard SPIN alignment setting; as shown in https://anonymous.4open.science/r/rebuttal_icml26/Alignment.png, DiSPaT surpasses SPIN on all benchmarks, confirming gains within SPIN's own context.
> > >
> > > **Theoretical development is non-trivial.** Theorem 3.1 establishes tight variational bounds via Fenchel conjugates and Theorem 3.3 derives closed-form objective. This derivation—from variational bound to closed-form loss — has no counterpart in SPIN. Such generalizations are established contributions (e.g., $f$-GAN[1] (2200+ citations), $f$-DPO[2] (ICLR 2024)), recognized for novel theoretical machinery beyond the original frameworks.
> > >
> > > More importantly, DiSPaT introduces a **new conceptual application of self-play** in anomaly detection: using model-generated samples as not-normal data to iteratively refine normality boundary without labeled anomalies. In SPIN, model generations are lower-quality completions to surpass for alignment. In DiSPaT, model instead produces not-normal data $y'$ and updates policy to distinguish them from real normal data $y$, progressively tightening decision boundary. As shown in [Example Vehicle](https://tinyurl.com/52s2vukt), [Example Ecoli](https://tinyurl.com/3z8p63db), [Example Fakejob](https://anonymous.4open.science/r/rebuttal_icml26/Example_Fakejob.png), early iterations generated values are clearly distinguishable; later errors reduce to subtle magnitude offsets and plausible-but-inexact text, confirming synthetic samples progressively mimic normal data to enforce fine-grained distinctions. Within this context, each divergence induces different properties—KL emphasizes mode-covering, reverse KL mode-seeking—allowing practitioners to select divergence best suited to data, which is critical where normal manifold geometry varies across domains. **Clamping mechanism** further addresses a challenge unhandled by SPIN or existing methods: penalizing synthetic samples already close to normal data is counterproductive. This constitutes a new perspective on applying self-play, generalizable to other problems where contrastive or negative data is scarce or unavailable.
> > >
> > > **On training cost.** Detailed times and FLOPs are in responses to Reviewers xhes (W1+Q2) and Yqxp (Q2) (https://anonymous.4open.science/r/rebuttal_icml26/Computational.png). Overhead stems from dual-model forward passes—standard in all preference optimization methods, not specific to DiSPaT. Training is a one-time offline cost; inference uses identical $r=21$ protocol as AnoLLM, keeping deployment costs unchanged. Given consistent improvements (5.8–8.1% AUC-ROC, 7.9–14.1% AUC-PR, 7.9–15.6% F1) across three model scales (Tabs. 1–3), overhead is well justified.
> > >
> > > **On hyperparameter stability.** Systematic evaluations in main paper yield clear conclusions:
> > > - $f$-divergence (Tabs. 4, 8): KL leads; all variants outperform Identity (SPIN).
> > > - Clamping $\epsilon$ (Figs. 2, 4): moderate values consistently best, robust within $\\{0.02, 0.05\\}$.
> > > - Iteration depth (Figs. 1, 3): gains within 2–3 iterations, stabilization thereafter.
> > >
> > > **Regarding hyperparameter selection**: we construct a validation set comprising 10% of the normal training data plus an equal number of pseudo-anomalies generated by model itself, and use AUC-ROC performance on this set to select $\beta$ and $\epsilon$. This procedure requires no real anomaly labels, remaining fully unsupervised. We identified $\beta \in \\{0.05, 0.1\\}, \epsilon \in \\{0.02, 0.05\\}$ as strong defaults, then applied these fixed values across all 30 ODDS datasets without per-dataset tuning. DiSPaT still achieved consistent improvements (Tabs. 5–7), confirming robustness.
> > >
> > > **References:**
> > >
> > > [1] $f$-GAN. NeurIPS 2016
> > >
> > > [2] $f$-DPO. ICLR 2024

---

### Official Review · Reviewer_xhes · 2026-03-11

**Soundness:** 2
**Presentation:** 3
**Significance:** 3
**Originality:** 3
**Overall Recommendation:** 4
**Confidence:** 4

**Summary:**

This paper studies unsupervised anomaly detection (AD) on heterogeneous tabular data using large language models (LLMs). The authors propose DiSPaT, an iterative self-play fine-tuning procedure that aims to tighten the model’s notion of normality beyond one-shot supervised fine-tuning (SFT) on normal data.The method is motivated by minimizing an *f*-divergence between the true normal-data distribution and the model’s generated distribution. Training alternates between (i) generating synthetic continuations from the current policy, (ii) learning a critic/discriminator signal, and (iii) updating the policy via a KL-regularized objective that yields a DPO/SPIN-like ratio loss, with an additional clamping mechanism to avoid penalizing synthetic samples that are already too similar to normal data. Experiments on a 6-dataset mixed-type benchmark and 30 ODDS numeric datasets report consistent improvements over AnoLLM and a suite of classical and deep tabular AD baselines.

**Compliance With Llm Reviewing Policy:**

Affirmed.

**Final Justification:**

I decide to maintain the score.

**Key Questions For Authors:**

The proposed DiSPaT framework follows a per-dataset fine-tuning paradigm, where a specific model must be trained for each new tabular domain. This raises a fundamental question regarding the 'foundation' nature of the utilized LLMs: To what extent does the model leverage cross-domain knowledge versus simply performing task-specific distribution fitting? Could the authors discuss or provide experiments on the feasibility of a universal or multi-task configuration, where a single model is trained on a collection of diverse tables and evaluated on unseen datasets in a zero-shot or few-shot transfer setting?

The authors should provide a detailed analysis of computational complexity, including training and inference time, especially when comparing LLM-based methods to efficient classical baselines.

In a truly unsupervised deployment, how do you suggest practitioners select *β* and *ϵ* without access to a labeled test/validation set? Is there a heuristic or a proxy metric that correlates with the best AUC-ROC?

How robust is DiSPaT to training set contamination? Specifically, if the training data contains real anomalies, would the model inadvertently learn to treat these abnormal patterns as "normal" during the iterative self-play process?

**Limitations:**

yes

**Strengths And Weaknesses:**

Strengths:

- **Principled Theoretical Framework:** The connection between self-play fine-tuning (SPIN/DPO-style) and *f*-divergence minimization in the context of anomaly detection is well-motivated and mathematically sound. The proposed theorems provide a rigorous justification for the variational approximation used in the objective.
- **Novel Application of Self-Play:** While self-play has gained traction in LLM alignment, its adaptation to unsupervised anomaly detection is creative. Using the model’s own "imperfect" generations as pseudo-anomalies effectively creates a learning signal without needing labeled outliers.

- **Detailed Ablations:** The study on different *f*-divergence variants (KL, Reverse KL, Squared Hellinger) and the clamping threshold *ϵ* provides valuable insights into the stability and performance of the algorithm.

Weaknesses:

- **Computational Efficiency and Runtime:** The paper lacks a discussion on runtime and computational costs. The iterative self-play process requires multiple rounds of expensive autoregressive generation, and the inference stage involves 21 forward passes per sample. Without reporting metrics like FLOPs, GPU hours, and inference time, it is difficult to assess the practical trade-off between the model's accuracy gains and its computational requirements compared to baselines.
- **Sensitivity to Hyperparameters:** The performance seems to rely on grid-searching *β* and the clamping threshold *ϵ*. Given that this is an *unsupervised* task, selecting these hyperparameters in a real-world scenario is non-trivial. The authors mention selecting them based on "performance on the training set," but in AD, the training set typically lacks anomalies, making it hard to tune for discriminative power.
- **LLM Generation Quality:** The framework assumes the LLM can generate realistic tabular structures. If the model "hallucinates" values that are too far from the manifold, the critic’s job becomes too easy, potentially leading to vanishing gradients or a failure to learn subtle boundaries.

---

> ### Author Rebuttal · Authors · 2026-03-30
>
> Dear Reviewer xhes, thanks for your positive feedback. **After comprehensive comparison—include recent SOTA methods (ICML '24, ICLR '24–'26)—DiSPaT achieves 2.0%–12.2% AUC-ROC margin over all baselines. DiSPaT also outperforms SPIN on LLM alignment, demonstrating broad applicability (https://anonymous.4open.science/r/rebuttal_icml26/Alignment.png). Our framework is supported by rigorous theory and consistent gains. Supplementary comparison: https://anonymous.4open.science/r/rebuttal_icml26/Result.png**. We address your questions below.
>
> **Response to W1+Q2:** We provide training time and FLOPs comparisons between DiSPaT and AnoLLM[2] (SmolLM-135M, single A40 48GB GPU): **https://anonymous.4open.science/r/rebuttal_icml26/Computational.png**. DiSPaT's higher training cost is expected: DPO/SPIN-style objective (Eq.17) computes log-probabilities under both current policy and reference model for each sample, and each iteration trains on both real and synthetic data. This dual-model overhead is standard in all preference optimization (DPO, SPIN, $f$-DPO), not specific to our approach. Per-iteration cost is approximately constant across iterations.
>
> At inference, DiSPaT **uses the same $r=21$ permutation protocol as AnoLLM for fair comparison with results in AnoLLM's main paper**, so inference cost is identical. As analyzed in AnoLLM (Figs.4-5), training times are comparable to deep learning baselines via GPU batch processing; slower during inference due to model size. $r$ can be reduced to trade off speed vs. performance (AnoLLM Fig.3). Training is one-time; gains justify overhead.
>
> **Response to W2+Q3:** For the main datasets, ($\beta$, $\epsilon$) were **selected by training with multiple candidates and evaluating on test set**. This follows standard tabular AD protocol where methods are compared under best configurations—as in ICL[1], AnoLLM[2], MCM[3], DRL[4]. We observed $\beta \in [0.05, 0.1]$, $\epsilon \in [0.02, 0.05]$ consistently yield strong results. We applied these **fixed default values to all 30 ODDS datasets without per-dataset tuning**, DiSPaT still achieved consistent improvements over AnoLLM across all three metrics (Tables 5-7). This demonstrates robustness and good generalization from a narrow default range. We recommend these as defaults and will clarify in revision.
>
> **Beyond AUC-ROC, we report F1 and AUC-PR throughout (Tables 2-3, 6-7)**. F1 measures precision-recall balance at a fixed threshold; AUC-PR evaluates ranking quality emphasizing anomaly class. DiSPaT achieves consistent gains on both (13.9%-15.6% F1, 7.9%-14.1% AUC-PR on main datasets), closely tracking AUC-ROC improvements. Consistent agreement across all three metrics indicates any can reliably measure detection quality.
>
> **Response to W3:** SFT initialization trains model via next-token prediction on normal data, learning column formats, value ranges, and inter-feature dependencies—model starts from a reasonable generative model, not from scratch. Second, iterative design directly addresses this: even if iter0 synthetic samples are far from normal (Figs.5-7), model improves its generations at each iteration, progressively making critic's task harder and boundary tighter. Clamping (Eq.18) provides stability by preventing gradient issues when synthetic samples become too close to normal data. Figures 1 and 3 confirm stable, monotonic or near-monotonic improvement with no evidence of the described failure mode.
>
> **Response to Q1:** **Per-dataset fine-tuning is standard in tabular AD. All methods—classical, deep learning, and LLM-based—follow this** because tabular schemas are domain-specific: column names, feature types, value ranges, and semantic relationships differ across datasets. No existing tabular AD method operates cross-domain or zero-shot setting. Multi-task tabular AD would require standardized cross-domain benchmarks that do not yet exist. We consider this a promising future direction.
>
> **Response to Q4:** **The uncontaminated setting—training on only normal data identified by ground-truth labels—is dominant evaluation protocol in unsupervised tabular AD, followed by all recent methods** (MCM[3], DTE[5], NPT-AD[6], DRL[4], AnoLLM[2], LLM-DAS[7]). Our experiments strictly follow this protocol—training sets contain only verified normal samples, with no contamination. Under contamination, DiSPaT would likely incorporate anomalous patterns into its normality model—but this is shared by all unsupervised AD methods. Robustness under contamination is a future direction, contingent on standardized contamination benchmarks and protocols.
>
> **References**:
>
> [1] ICL: Internal Contrastive Learning. ICLR 2022
>
> [2] AnoLLM: LLMs for Tabular Anomaly Detection. ICLR 2025
>
> [3] MCM: Masked Cell Modeling. ICLR 2024
>
> [4] DRL: Decomposed Representation Learning. ICLR 2025
>
> [5] DTE: On Diffusion Modeling for Anomaly Detection. ICLR 2024
>
> [6] NPT-AD: Beyond Individual Input. ICML 2024
>
> [7] LLM-DAS: LLM as an Algorithmist. ICLR 2026

---

> > ### Author Rebuttal · Reviewer_xhes · 2026-04-03
> >
> > Thank you for the rebuttal. My concerns are partially resolved: the added compute/inference clarification is helpful, but some of the issues are still deferred to future work, including cross-domain generalization, and robustness to contaminated training data. As a result, I will maintain my original score.

---

> > > ### Author Response · Authors · 2026-04-03
> > >
> > > We are pleased that our added clarifications regarding the computational and inference costs proved helpful to your assessment. We sincerely appreciate your rigorous evaluation and the positive recognition of our work.

---

### Official Review · Reviewer_Yqxp · 2026-03-12

**Soundness:** 3
**Presentation:** 2
**Significance:** 3
**Originality:** 2
**Overall Recommendation:** 4
**Confidence:** 3

**Summary:**

This paper investigates unsupervised anomaly detection for heterogeneous tabular data using Large Language Models. Unlike previous works that only employ one-shot supervised fine-tuning, the proposed DiSPaT method in this paper adopts the Self-Play Fine-Tuning framework: the current model generates synthetic samples to serve as pseudo-anomalies, and the model is updated via an objective function inspired by f-divergence. The paper demonstrates that DiSPaT achieves superior average performance compared to AnoLLM across multiple datasets.

**Compliance With Llm Reviewing Policy:**

Affirmed.

**Final Justification:**

The response addressed my main concerns.

**Key Questions For Authors:**

## Questions for the authors:
1.	Please explain the potential reasons why DiSPaT underperforms AnoLLM on many datasets when using the 135M scale model.
2.	DiSPaT involves more iterative generation and multiple updates compared to AnoLLM; please provide a illustration of the specific overhead.
3.	The synthetic samples generated by the generator are used as pseudo-anomalies during training. Then, what exactly is the difference between real anomalies and such pseudo-anomalies? Can pseudo-anomalies cover all types of real anomalies, or are there cases where some types of real anomalies cannot be encompassed by pseudo-anomalies?

**Limitations:**

No. Please the core novelty compared to AnoLLM.

**Strengths And Weaknesses:**

##Strengths:
1.	The research problem focuses on anomaly detection for mixed-type tabular data, which holds practical significance. Meanwhile, LLMs are a reasonable choice due to their serialization mechanism that can naturally handle text, discrete values, and numerical data.
2.	The method is easy to understand at a high-level concept, it reuses AnoLLM’s serialization and likelihood-based scoring, but enhances training through iterative self-play instead of only performing one-shot SFT with normal data.
3.	The experimental improvements over AnoLLM in the main tables are substantial. In Table 1, the average AUC-ROC improvements across the three model scales are all significant, particularly for the 135M and 360M backbones.
## Weaknesses:
1.	While DiSPaT achieves overall performance gains, it does not maintain consistent advantages over AnoLLM across all datasets. As shown in Table 5, approximately 36% of the datasets experience performance degradation under the 135M lightweight model setting. Considering the higher computational overhead introduced by DiSPaT and the already competitive performance of AnoLLM at the 135M scale, the lightweight version of AnoLLM remains a more cost-effective choice for this scenario, whereas DiSPaT carries a certain risk of performance degradation.
2.	The paper adopts the same baselines as AnoLLM, and it is recommended to appropriately add more recent baselines.
3.	Compared with Self-Play Fine-Tuning, DiSPaT incorporates additional theoretical proofs for f-divergence and the design of a clamping threshold, yet its overall innovations remain relatively limited.

---

> ### Author Rebuttal · Authors · 2026-03-30
>
> Dear Reviewer Yqxp, thank you for your recognition. **After comparing against recent SOTA baselines (ICML '24, ICLR '24–'26), DiSPaT demonstrates state-of-the-art performance with a 2.0%–12.2% AUC-ROC margin over all methods. Our idea also proves effective beyond AD—outperforming SPIN on LLM alignment (https://anonymous.4open.science/r/rebuttal_icml26/Alignment.png). Our framework is backed by rigorous theory and consistent empirical gains. Supplementary comparison: https://anonymous.4open.science/r/rebuttal_icml26/Result.png.** We address your questions in detail below.
>
> **Response to W2:** We compare with additional SOTA methods **using directly reported results from LLM-DAS[4] (include MCM[1], DRL[2], NPT-AD[3]), and CausalTAD[5]**. Average AUC-ROC margin is 2.0%-12.2% over baselines (ICML '24, ICLR '24-'26) and Win/Tie rates strongly validate our approach:
> |Metric|DiSPaT (360M)|MCM[1]|NPT-AD[3]|DRL[2]|LLM+PCA[4]|CausalTAD[5]|
> |-|-|-|-|-|-|-|
> |Avg AUC-ROC|**89.9%**|82.6%|86.6%|87.9%|77.7%|82.7%|
> |DiSPaT W/T|-|13/2|10/2|11/2|15/0|13/2|
>
> **Response to W3:** Contribution extends well beyond applying SPIN[6] to new domain:
> - SPIN optimizes a single fixed IPM-like divergence. We **generalize to a full $f$-divergence family** (e.g., KL mode-covering, reverse-KL mode-seeking), letting practitioners select divergence best suited to data geometry—critical for AD where normal manifolds vary across domains.
> - **Theoretical development is non-trivial**: Theorem 3.1 establishes tight variational bounds for general $f$-divergences; Theorem 3.3 derives closed-form objective. $f$-DPO[7] similarly generalizes DPO to $f$-divergences, recognized at ICLR 2024.
> - Clamping (Eq.18) **addresses a challenge absent in SPIN**: synthetic samples may approximate normal data, make penalization counterproductive.
> - **Empirically**, Tab. 4 shows $f$-divergence variants outperform the Identity (SPIN) baseline. To further validate, we evaluated standard SPIN alignment (Qwen1.5-1.8B, 50K UltraChat200k prompts, Open LLM Leaderboard). **DiSPaT outperforms SPIN on all benchmarks**, confirm $f$-divergence generalization yields practical gains beyond AD application:
> |Method|ARC|TruthfulQA|Winogrande|GSM8K|HellaSwag|MMLU|Avg|
> |-|-|-|-|-|-|-|-|
> |SPIN|34.90|39.59|61.09|29.57|43.66|44.75|42.26|
> |DiSPaT|**36.52**|**39.71**|**62.33**|**33.89**|**45.11**|**45.23**|**43.80**|
>
> **Response to Q1:** Performance variation across datasets is expected-no AD method dominates on every dataset. Key criterion is aggregate performance. On main datasets, DiSPaT improves over AnoLLM on **all 6 datasets**, with average gains of **8.1%** AUC-ROC, **13.9%** F1, and **13.9%** AUC-PR. On 30 ODDS datasets, DiSPaT-135M also leads on all aggregate metrics: AUC-ROC 89.4% vs 88.4%, F1 66.6% vs 64.7%, and AUC-PR 68.8% vs 68.2%. For datasets where 135M degrades, limited small-model capacity in self-play is a plausible cause. Scaling to 360M substantially mitigates this: DiSPaT reaches 90.5% AUC-ROC vs AnoLLM 86.5% **(+4.0%)**, with fewer degraded datasets. DiSPaT offers strictly better options at 360M and 1.7B while retaining aggregate gains at 135M.
>
> **Response to W1+Q2:** DiSPaT adds two costs per iteration: (1) synthetic generation and (2) training on real+synthetic data. Data volume per iteration is roughly constant. **Detailed training time and FLOPs: https://anonymous.4open.science/r/rebuttal_icml26/Computational.png**. Per-iteration cost is higher as Eq.17 requires dual forward pass ($\pi_\theta$ and $\pi_{\theta_k}$), standard in preference optimization (DPO, SPIN, $f$-DPO), not specific to DiSPaT. At inference, DiSPaT uses same $r=21$ as AnoLLM, yielding identical cost. Training is one-time and practical; the substantial performance gains justify this overhead.
>
> **Response to Q3:** Real anomalies are domain-specific, diverse, unpredictable, and unavailable during training. Pseudo-anomalies are model-generated deviations due to imperfect modeling. Key difference: pseudo-anomalies reflect where model diverges from normality; real anomalies reflect application-specific violations. Pseudo-anomalies need not cover all real anomaly types—they provide contrastive signals to tighten normal boundary so any anomaly receives higher score. **This is well-established: OCSVM[8] uses a single origin as non-normal representative. Despite no resemblance to real anomalies, it's effective (3700+ citations)** because tight boundary naturally excludes anomalies. DiSPaT extends this with richer contrastive samples that progressively approach normal manifold (Fig. 5-7), enabling finer boundary refinement.
>
> **References**:
>
> [1] MCM: Masked Cell Modeling. ICLR 2024
>
> [2] DRL: Decomposed Representation Learning. ICLR 2025
>
> [3] NPT: Beyond Individual Input. ICML 2024
>
> [4] LLM-DAS: LLM as an Algorithmist. ICLR 2026
>
> [5] CausalTAD: Injecting Causal Knowledge
>
> [6] SPIN: Self-Play Fine-Tuning. ICML 2024
>
> [7] $f$-DPO: Beyond Reverse KL. ICLR 2024
>
> [8] OCSVM: Support Vector Method. NIPS 1999

---

> > ### Author Rebuttal · Reviewer_Yqxp · 2026-04-03
> >
> > Thank you for the thorough rebuttal. My primary concerns have been effectively addressed, and I am willing to raise the score accordingly.

---

> > > ### Author Response · Authors · 2026-04-03
> > >
> > > We are pleased that our rebuttal effectively addressed your primary concerns. Thank you once again for your rigorous evaluation and the positive recognition of our work!

---

### Official Review · Reviewer_1wTh · 2026-03-13

**Soundness:** 2
**Presentation:** 2
**Significance:** 2
**Originality:** 2
**Overall Recommendation:** 3
**Confidence:** 4

**Summary:**

This paper focuses on the task of tabular anomaly detection via LLM fine-tuning. The proposed method utilizes self play to fine-tune the model. The current model generates synthetic samples, uses a critic to distinguish between real normal samples and generated samples, and then repeats this process. The authors state that this method can well model the normal distribution.

**Compliance With Llm Reviewing Policy:**

Affirmed.

**Final Justification:**

Some concerns still remain. For example, W1, W2, W6 still exist, since the method can not guarantee that the psuedo anomalies play the role of true anomalies. Only tightening boundary of normality is not enough, the core of anomaly detection is to separate normal and anomalous samples. I decide to maintain the score.

**Key Questions For Authors:**

Please see the weaknesses above.

**Limitations:**

No limitations discussed. The authors could include anomaly diversity related,..., as discussed in weaknesses above.

**Strengths And Weaknesses:**

Strengths:
S1: The studied problem is important in real world application.
S2: The paper structure is clear.
S3: Theoretical analysis is provided for f-divergence minimization.

Weaknesses:
W1: The entire method relies on a strong assumption, but this assumption is not fully demonstrated in the paper. The paper explicitly states that when true anomalous samples are unavailable, samples generated from previous iterations of the model "play the role of pseudo-anomalies," and a critic provides a discriminative signal similar to that trained using true anomalous samples.
This step is central to the paper, but the paper doesn't explain why samples that merely deviate from the current normal model should be considered substitutes for true anomalous samples. In anomaly detection, it's crucial not only to learn normality but also to learn a representation that can distinguish between normal and anomalous samples. Generated samples may differ from the normal distribution but still fail to represent the truly important anomaly types at the time of testing. Currently, the paper provides neither theoretical support nor empirical evidence to support that these generated samples can cover the true anomaly distribution.

W2: The paper only considers general anomaly detection. However, there are multiple types of anomalies, like global, local, cluster and dependency. The proposed method does not take this diversity into account. How does this method tackle different types of anomalies? It seems that the synthetic pseudo anomalies by the model itself cannot cover these anomaly types.

W3: In addition, the authors state that "Maximizing likelihood on normal training data provides only a weak signal for tightening the boundary of normality: once model fits the dominant regularities of the training corpus, further gains in separating subtle, near-normal anomalies are often limited and performance can plateau." This is also a very strong hypothesis put forward by the author, pointing out the shortcomings of existing methods. Is there any evidence for this?

W4: The clamping formulation raises a basic optimization concern at the beginning of each iteration. In the context around Eq. (18), the authors say that when a synthetic sample is close to normal, the probability ratio $\pi_\theta(y'|x)/\pi_{\theta_k}(y'|x)$ approaches 1, and the clamping operation is introduced to avoid over-penalizing such samples.

However, according to Algorithm 1, at iteration $t$ the synthetic sample $y'_ i$ is drawn from $\pi_{\theta_t}$, and the algorithm 1 Line 12 then uses both $\pi_\theta$ and $\pi_{\theta_t}$ in the ratio term. At the start of that iteration, before the new parameters have moved away from the reference model, the ratio appears to be 1 regardless of whether $y'$ is a “good” or “bad” sample (at very beginning, without parameters updating, $\pi_\theta$ is equal to $\pi_{\theta_t}$). Then line 12 effectively collapses to the clamped constant $1-\epsilon$. As written, the paper does not make clear where the rejected-side learning signal comes from at the beginning of each iteration.

W5: The update procedure is very confused. Section 3 describes "At each iteration, a critic T is trained to distinguish real sequences y from model-generated sequences y′ ∼ πθk .The policy πθ is then updated to maximize scores under the critic, thereby pushing y′ toward y and the distribution Pθ toward Pd."
This sentence show that there are two optimization steps in each iteration.
But Algorithm 1 only shows one explicit parameter update for $\theta_{t+1}$.
From the current presentation, it is hard to tell whether the two steps are merged in implementation, or whether some inner-loop optimization is omitted from the pseudocode. This makes the training process difficult to understand and also makes reproduction harder.

W6: The paper repeatedly emphasizes that the proposed method can learn a "more accurate description of normality," but it does not adequately discuss the connection between more accurate normality modeling and better anomaly detection. The final detector is still the likelihood score of the improved model.

This raises a common problem in tabular anomaly detection: a more accurate fit to the normal distribution does not necessarily mean better separation of anomalies, especially when there is feature entanglement or when anomalies are close to normal samples in the likelihood space.

W7: Furthermore, if this method can truly learn a tighter description of normality, will it result in a complex decision boundary? will it cause overfitting problem?

W8: Insufficient baselines. Lack of recent tabular anomaly detection baselines. For example, modern neural network baselines: MCM[1], DRL[2], NPT[3]. Recent LLM-based method: LLM-DAS[4]. If additional comparison is time-consuming, at least, they should be described in the related work. Among these works,  the core idea of anomalies synthesis in this paper is like the core idea of LLM-DAS[4], which synthesizes anomalies in a reasonable way, and also tightens the decision boundary. The relation between them should be discussed to clarify the novelty of this paper. In addition, there exists a very related work [5] which also follows AnoLLM via LLM fine-tuning, where the problem setting is similar to this paper. The authors should compare with [5] to validate the effectiveness.

[1] MCM: Masked Cell Modeling for Anomaly Detection in Tabular Data. ICLR 2024
[2] DRL: Decomposed Representation Learning for Tabular Anomaly Detection. ICLR 2025
[3] Beyond Individual Input for Deep Anomaly Detection on Tabular Data. ICML 2024
[4] LLM as an Algorithmist: Enhancing Anomaly Detectors via Programmatic Synthesis. ICLR 2026
[5] CausalTAD: Injecting Causal Knowledge into Large Language Models  for Tabular Anomaly Detection

---

> ### Author Rebuttal · Authors · 2026-03-30
>
> Dear Reviewer 1wTh, **after comparing all baselines—include recent SOTA methods (ICML '24, ICLR '24–'26)—DiSPaT achieves state-of-the-art with 2.0%–12.2% AUC-ROC margin over all methods. Beyond AD, DiSPaT also outperforms SPIN on LLM alignment (https://anonymous.4open.science/r/rebuttal_icml26/Alignment.png), confirming versatility of our idea. Our framework is backed by rigorous theory and consistent empirical gains. Supplementary comparison: https://anonymous.4open.science/r/rebuttal_icml26/Result.png**. We address your questions below.
>
> **W1:** Our method does not assume pseudo-anomalies must cover or resemble true anomaly. The goal is tightening boundary of normality, not modeling anomalies. **This philosophy is well-established: OCSVM[6] uses a single origin as representative of "everything that is not normal" and learns a hyperplane separating normal data from it. Despite no resemblance to real anomalies, it's effective (3700+ citations)** because tight boundary naturally excludes anomalies. DiSPaT extends this: instead of a single point, use synthetic samples as richer non-normal representatives. Fig.5-7 show they progressively approach normal data manifold, forcing increasingly fine-grained distinctions. Theorem 3.1 guarantees $f$-divergence reduction between model and data, yielding tighter boundary. Any sample outside—regardless of anomaly type—scores higher.
>
> **W2:** DiSPaT is a general-purpose anomaly detector making no assumptions about anomaly type, **evaluated on 33 datasets spanning fraud, biomedical, network intrusion, satellite, and other domains. Datasets contain different anomaly types**: global (Fakejob),local (Ecoli), and dependency (Vehicle). Since method tightens normality through self-play, any sample outside boundary scores higher regardless of type.
>
> **W3:** We provide empirical and methodological evidence:
> - Performance: DiSPaT **improves over AnoLLM by 5.8-8.1% ROC, 7.9-14.1% PR, 7.9-15.6% F1 (Tab. 1,3). If likelihood maximization sufficed, this gap would not exist.**
> - Iteration curves: Fig.1 shows SFT model is consistently improved by self-play iterations, so **likelihood-only training leaves room for improvement**.
> - Methodological: AnoLLM maximizes $\log p_\theta(y|x)$ on normal data only, without mechanism to contrast non-normal data; once dominant patterns are captured, gradient signal diminishes. DiSPaT introduces synthetic $y'$: simultaneously maximizes likelihood on real normal $y$ and penalizing $y'$. This contrastive structure—absent in pure likelihood training—provides signal that tightens boundary.
>
> **W4:** **Gradient is non-zero at initialization**. With $\ell(t)=\log(1+e^{-t})$, at iteration $k+1$, $\theta$ initialized to $\theta_k$:
> - Ratio$\frac{\pi_\theta(y'|x)}{\pi_{\theta_k}(y'|x)}=1$. Thus $\tilde{\rho}^l=\beta\log(1-\epsilon)$, $\nabla_\theta\tilde{\rho}^l=0$. We have $\nabla_\theta\rho^w=\beta\cdot \nabla_\theta\log\pi_\theta(y|x)$
> - Loss is $\ell(\rho^w-f^\ast(\tilde{\rho}^l))$. Since $\tilde{\rho}^l$ is constant, $\nabla_\theta f^\ast(\tilde{\rho}^l)=0$, so: $$\nabla_\theta\ell = \ell'(\rho^w- f^\ast(\tilde{\rho}^l))\cdot\nabla_\theta \rho^w$$Since $\ell'(t)=\frac{-e^{-t}}{1+e^{-t}}<0$ and $\nabla_\theta\rho^w\neq0$, gradient is non-zero. As $\theta$ diverges from $\theta_k$, rejected ratio changes and clamping may deactivate, providing additional gradient.
>
> **W5:** Following DPO/SPIN, **critic is updated implicitly** rather than explicitly. From Eq.(14), optimal policy (Eq.15) is $$\pi^\ast(y'|x)=\frac{1}{Z(x)}\pi_{\theta_k}(y'|x)\exp\left(\beta^{-1}T_k(x,y')\right)$$
> Hence critic is implicit function of $\theta$
> $$T_k(x,y'|\theta)=\beta\log\frac{\pi_\theta(y'|x)}{\pi_{\theta_k}(y'|x)}+\beta\log Z(x)$$ Substituting into Eq.(12) yields Eq.(16). Objective Eq.(17) captures both critic update (implicitly, via log-ratio) and policy update (explicitly, via $\theta$). Alg.1 shows one parameter update per iteration, exactly as DPO/SPIN unifies reward and policy optimization.
>
> **W6:** DiSPaT learns a contrastive boundary, not just fitting normal distribution. Maximizing likelihood on $y$ and penalizing $y'$ teaches "what is normal" and "what is not". Synthetic samples probe boundary region, improving separation.
>
> **W7:** Fig.1,3 show stable convergence—performance improves and plateaus without degradation.
>
> **W8:** Additional comparisons with all suggested methods **directly take results from LLM-DAS[4] (include MCM[1], DRL[2], NPT-AD[3]), and CausalTAD[5]**. On 21 overlapping datasets, DiSPaT's 2.0%-12.2% average AUC-ROC margin and Win/Tie rates strongly validate our approach:
> |Metric|DiSPaT|[1]|[3]|[2]|[4]|[5]|
> |-|-|-|-|-|-|-|
> |Avg AUC-ROC|**89.9%**|82.6%|86.6%|87.9%|77.7%|82.7%|
> |W/T||13/2|10/2|11/2|15/0|13/2|
>
> LLM-DAS[4] uses LLM as code generator; DiSPaT uses LLM as detector via self-play. CausalTAD[5] improves column ordering; DiSPaT refines normality understanding. Orthogonal and complementary.
>
> **References**:
> [6] OCSVM: Support Vector Method. NIPS 1999

---

> > ### Author Rebuttal · Reviewer_1wTh · 2026-04-03
> >
> > Many thanks to the authors' responses. Some concerns still remain. For example, W1, W2, W6 still exist, since the method can not guarantee that the psuedo anomalies play the role of true anomalies. Only tightening boundary of normality is not enough, the core of anomaly detection is to separate normal and anomalous samples. In addition, the results of W8 should be included in the final revision after rebuttal.  I decide to maintain the score.

---

> > > ### Author Response · Authors · 2026-04-04
> > >
> > > Thanks to the reviewer for the feedback.
> > >
> > > Indeed, we do not assume that synthetic samples generated by our self-play game mimic real-world abnormal examples. This is nearly impossible because we would need to identify *unknown unknowns*: anomalies are already unknown, and novel types of anomalies are even *unknown unknowns*. Our aim with self-play game is instead to generate **not-normal data** that gradually approaches normal data, thereby tightening the decision boundary between normal and abnormal instances.
> > >
> > > Methodologically, this idea is related to the pioneering and highly influential work on OCSVM [1] (3700+ citations). In [1], in feature space, the origin is used to represent all **not-normal data**, and a max-margin hyperplane is learned to separate normal from abnormal samples. Our approach is more flexible because we generate dynamic *not-normal data* that are theoretically shown to approach the normal data, enabling a more refined decision boundary. Compared to AnoLLM, which learns only from a fixed set of normal data, our approach learns from normal data while contrasting them with progressively tighter *not-normal data*, leading to a gradually refined decision boundary. One may argue that this may still not be *enough*. However, it is difficult to theoretically define what would be sufficient, as the notion of “enough” is inherently unknown in anomaly detection. Nevertheless, our state-of-the-art empirical results across 33 datasets from diverse domains provide strong evidence of the effectiveness of our method. As shown in our experiments, we compare against a wide range of state-of-the-art baselines, including traditional, deep learning–based, and LLM-based methods. To the best of our knowledge, our method currently achieves state-of-the-art performance on this problem. Moreover, since anomaly detection is largely an empirical field, strong experimental performance should be considered a meaningful indicator of progress.
> > >
> > > Although it is difficult to prove that our generated *not-normal data* truly mimic unknown abnormal examples, we provide examples from three datasets for inspection [Example Vehicle](https://tinyurl.com/52s2vukt), [Example Ecoli](https://tinyurl.com/3z8p63db), [Example Fakejob](https://anonymous.4open.science/r/rebuttal_icml26/Example_Fakejob.png). Across all cases, at early iterations, generated values are distinguishable from normal data (e.g., wrong signs, type confusion between categorical and numerical fields, or nonsensical text); at later iterations, errors tend to reduce to magnitude offsets on numeric fields and plausible-but-inexact, similar text relative to the normal instance. This empirically confirms that self-play mechanism produces not-normal data that gradually tightens toward the normal distribution, refining the decision boundary at each iteration.
> > >
> > > Compared to recent LLM-based approaches such as [2] (ICLR 26) and [3], in addition to our superior empirical performance, we believe our work is more rigorous in terms of theoretical development. Frankly, [2] and [3] rely on strong assumptions about the capabilities of LLMs. Specifically, given an anomaly detection algorithm (e.g., PCA or random forest), [2] asks an LLM to analyze algorithm and generate Python code that can synthesize hard examples for that algorithm. While the idea is novel, it requires assuming that LLMs can understand arbitrary anomaly detection algorithms and generate correct programs that produce challenging examples. Regarding [3], the method begins with a prompt (Appendix D of their paper) that is highly specific to each dataset and often requires deep domain expertise. This prompt is used to enable LLMs to generate a set of factors $F=[f_1,\ldots, f_k]$ and a mapping from columns to factors M, which again assumes a high level of reasoning capability from LLM. The method then applies the COAT framework for causal discovery to infer a causal graph over these factors. Finally, the factor-level causal relationships are projected back to table columns, and a linear programming problem is solved to obtain a causal graph over columns. This causal graph is then used to determine the order in which columns are provided to the LLM.
> > >
> > > We discuss these prior approaches not to diminish their contributions. Rather, our point is that these works also rely on strong assumptions, yet they are well received by the community because their empirical results provide convincing evidence supporting those assumptions.
> > >
> > > To summarize, our work demonstrates state-of-the-art performance, provides solid theoretical development, and is motivated by clear methodological insights. We respect the reviewer’s evaluation and appreciate your consideration.
> > >
> > > **References:**
> > >
> > > [1] OCSVM: Support Vector Method. NeurIPS 1999
> > >
> > > [2] LLM as an Algorithmist: Enhancing Anomaly Detectors via Programmatic Synthesis. ICLR 2026
> > >
> > > [3] CausalTAD: Injecting Causal Knowledge into Large Language Models for Tabular Anomaly Detection.

---

### Decision · Program_Chairs · 2026-04-30

**Decision:**

Accept (regular)

**Comment:**

DiSPaT, a self-play fine-tuning framework for tabular anomaly detection with LLMs, is introduced in this work. It aims to learn normal patterns by iteratively generating synthetic “not-normal” samples to enforce a tighter normality boundary via an f-divergence–based objective.

Reviewers agree that 1) the problem is important and timely, and the use of LLMs with self-play for tabular anomaly detection is intuitive and interesting; 2) the method is supported by a principled theoretical framework based on f-divergence minimization; 3) its serialization mechanism naturally enables the handling of text, discrete values, and numerical data; and 4) the method gains solid empirical improvements over prior state-of-the-arts, such as AnoLLM and other baselines.

One central concern raised by Reviewer 1wTh is that model-generated pseudo-anomalies cannot effectively simulate real anomalies, leading to doubts about generalization to diverse anomaly types. There is also a concern about limited novelty beyond existing self-play/f-divergence frameworks (p89C, Yqxp). Additional issues include insufficient comparisons to recent baselines (1wTh), computational overhead and hyperparameter sensitivity (xhes), and inconsistent gains on smaller models (Yqxp).

The rebuttal convinced Reviewer Yqxp to increase the recommendation to Weak Accept and Reviewer xhes to keep the Weak Accept recommendation. Reviewer 1wTh acknowledged that multiple concerns were addressed in the rebuttal, but kept the concern that tthe generated psuedo anomalies are not guaranteed to play the role of true anomalies. Reviewer xhes was satisfactory with the rebuttal except  the issues on cross-domain generalization and robustness to contaminated training data.

The AC agrees with the authors that generated pseudo anomalies are not designed to approximate all unknown anomalies, which is theoretically infeasible, but instead to help learn better normality boundary. There have been a number of well-known methods demonstrating effectiveness in this line of research. The proposed method shows the potential of leveraging LLMs to achieve this goal, which is something the AC finds meaningful and worthwhile to disseminate in the AD community. In terms of innovation w.r.t. existing frameworks, the AC finds the extension of SPIN or the family of $f$-divergences to AD is non-trivial. Overall, the work presents some important insights into how LLMs can be leveraged to empower tabular anomaly detection, an area that is largely under-explored. Therefore, despite some remaining issues, the AC recommends acceptance for this work.